# Single-molecule FRET unmasks structural subpopulations and crucial molecular events during FUS low-complexity domain phase separation

Ashish Joshi[1,2], Anuja Walimbe[1,2], Anamika Avni[1,3,4], Sandeep K. Rai[1,3], Lisha Arora[1,3], Snehasis Sarkar[1,2] & Samrat Mukhopadhyay [1,2,3] ✉

Biomolecular condensates formed via phase separation of proteins and nucleic acids are thought to be associated with a wide range of cellular functions and dysfunctions. We dissect critical molecular events associated with phase separation of an intrinsically disordered prion-like low-complexity domain of Fused in Sarcoma by performing single-molecule studies permitting us to access the wealth of molecular information that is skewed in conventional ensemble experiments. Our single-molecule FRET experiments reveal the coexistence of two conformationally distinct subpopulations in the monomeric form. Single-droplet single-molecule FRET studies coupled with fluorescence correlation spectroscopy, picosecond time-resolved fluorescence anisotropy, and vibrational Raman spectroscopy indicate that structural unwinding switches intramolecular interactions into intermolecular contacts allowing the formation of a dynamic network within condensates. A disease-related mutation introduces enhanced structural plasticity engendering greater interchain interactions that can accelerate pathological aggregation. Our findings provide key mechanistic underpinnings of sequence-encoded dynamically-controlled structural unzipping resulting in biological phase separation.

Living cells compartmentalize their biochemical components and processes using well-defined membrane-bounded organelles. A growing body of rapidly evolving research reveals that in addition to such conventional membrane-bounded organelles, cells contain non-canonical membraneless organelles that are thought to be formed via liquid–liquid phase separation of proteins, nucleic acids, and other biomolecules[1–10]. These membraneless compartments, also termed biomolecular condensates, include nucleolus, stress granules, P-bodies, Cajal bodies, nuclear speckles, and so on. These membraneless bodies are highly dynamic, liquid-like, regulatable, permeable, non-stoichiometric supramolecular assemblies involved in the spatio-temporal regulation of vital cellular processes including genome organization, RNA processing, signaling, transcription, stress regulation, immune response, and so forth[11–13]. Recent studies have established that intrinsically disordered proteins/regions (IDPs/IDRs) possessing low-complexity and prion-like domains are the key

[1]Centre for Protein Science, Design and Engineering, Indian Institute of Science Education and Research (IISER) Mohali, Sector 81, SAS Nagar, Mohali, Punjab 140306, India. [2]Department of Biological Sciences, Indian Institute of Science Education and Research (IISER) Mohali, Sector 81, SAS Nagar, Mohali, Punjab 140306, India. [3]Department of Chemical Sciences, Indian Institute of Science Education and Research (IISER) Mohali, Sector 81, SAS Nagar, Mohali, Punjab 140306, India. [4]Present address: Department of Integrative Structural and Computational Biology, The Scripps Research Institute, 10550 North Torrey Pines Road, La Jolla, CA 92037, USA. ✉e-mail: mukhopadhyay@iisermohali.ac.in

candidates for biological phase separation[7,8,14–16]. These studies revealed that the presence of low-sequence complexity promotes intrinsic disorder, conformational flexibility, structural heterogeneity, and multivalency that enable the polypeptide chains to participate in a multitude of ephemeral chain–chain interactions governing the making and breaking of noncovalent interactions on a characteristic timescale. These noncovalent intermolecular interactions involve electrostatic, hydrophobic, hydrogen bonding, dipole–dipole, π–π, and cation–π interactions and yield a highly dynamic liquid-like behavior of phase-separated biomolecular condensates[17–22]. Biomolecular condensate formation involves a density transition coupled to the percolation that results in the dense phase comprising a viscoelastic network fluid[4]. Such condensates comprising viscoelastic fluids can undergo aberrant liquid-to-solid phase transitions resulting in the maturation and hardening of these assemblies into solid-like aggregates that are associated with a wide range of neurodegenerative diseases[2,12–14,23].

An archetypal phase-separating protein, Fused in Sarcoma (FUS), is a highly abundant protein belonging to the FET (FUS, EWSR1, and TAF15) family of proteins. Liquid-like condensates of FUS are thought to play crucial roles in RNA processing, DNA damage repair, paraspeckle formation, miRNA biogenesis, and the formation of stress granules. On the contrary, solid-like aggregates of FUS are identified as pathological hallmarks of several neurodegenerative diseases, including amyotrophic lateral sclerosis (ALS) and frontotemporal dementia (FTD)[24–28]. FUS exhibits a multidomain architecture comprising an intrinsically disordered N-terminal domain and a partly structured C-terminal RNA-binding domain. The N-terminal domain contains a QSGY-rich prion-like low-complexity domain, whereas, the C-terminal RNA-binding domain (FUS-RBD) consists of an RNA-recognition motif (RRM), two RGG-rich stretches, a zinc finger domain, and a short nuclear localization signal (NLS) (Fig. 1a). The N-terminal, intrinsically disordered, prion-like low-complexity domain, termed FUS-LC, has been identified as the major driver of self-assembly into liquid-like condensates, hydrogels, and solid-like aggregates (Fig. 1b)[25–29]. Previous studies have established that FUS-LC serves as a model prion-like system to investigate the fundamental biophysical principles of phase separation and maturation. Structural characterizations have indicated that FUS-LC remains intrinsically disordered in both monomeric dispersed and condensed phases[25,30–37]. However, the complex interplay of the key molecular determinants and the crucial molecular events that critically govern the course of macromolecular phase separation of FUS-LC remain elusive. The key question of how sequence-encoded conformational plasticity, structural distributions, and chain dynamics control weak, multivalent, transient intermolecular interactions resulting in the formation of liquid-like condensates is of paramount importance both in normal cell physiology and disease biology.

In this work, we elucidate the structural heterogeneity, distributions, and interconversion dynamics of FUS-LC using single-molecule experiments that permit us to monitor the conformational states in a molecule-by-molecule manner. Such single-molecule experiments offer a powerful approach to accessing the incredible wealth of molecular information that is normally skewed in conventional ensemble-averaged experiments[38–44]. These single-molecule studies allow us to interrogate one molecule at a time and directly capture the hidden conformational states, characteristic conformational fluctuations, and interconversion dynamics. We utilized highly sensitive single-molecule FRET (Förster resonance energy transfer) using FUS-LC constructs site-specifically and orthogonally labeled with a donor-acceptor pair (Fig. 1c) that allowed us to detect and characterize structurally distinct states and conformational distributions within structurally heterogeneous populations in the monomeric dispersed phase and the protein-rich condensed phase. Our single-molecule FRET studies varying the inter-residue distance in conjunction with fluorescence correlation spectroscopy (FCS), picosecond time-resolved fluorescence anisotropy, and vibrational Raman spectroscopy within individual phase-separated condensates permitted us to dissect the conformational shapeshifting events associated with phase separation of FUS-LC. We also elucidated the impact of a clinically relevant pathological mutation on the conformational distribution and dynamics that alters the phase behavior of the low-complexity domain.

## Results

### Single-droplet FRET imaging hints at long-range intramolecular interactions in the condensed phase

FUS-LC comprising 163 residues exhibits a near-uniform distribution of amino acids, serine (S), tyrosine (Y), glycine (G), and glutamine (Q), and is characterized by a low net charge and low mean hydrophobicity (Fig. 1b, d). We began with the bioinformatics characterization using PONDR[45], which confirmed the presence of intrinsic disorder in FUS-LC (Fig. 1e), and CIDER[46] (Classification of Intrinsically Disordered Ensemble Relationships) which provides various distinctive features of IDPs built on the sequence composition. Using CIDER, we obtained the NCPR (net charge per residue) value from the NCPR plot and the diagram of states which illustrates the distribution of positive and negative charges throughout the sequence. IDPs often possess a sequence composition that is characterized by a low mean hydrophobicity and a high net charge[47]. However, FUS-LC carries a low net charge with an NCPR value <0.25; such a polypeptide can exist as compact globular ensembles in contrast to the well-solvated expanded conformations exhibited by charged IDPs[46,48]. Based on the charge composition, FUS-LC is predicted to adopt a compact[49] or tadpole-like[46] structure, as also evident from the diagram of states, which predicts IDP conformations based on the fraction of positively and negatively charged residues within the sequence (Fig. 1f). To experimentally validate, we recombinantly expressed and purified FUS-LC and performed circular dichroism (CD) spectroscopic measurements of monomeric FUS-LC which exhibited a characteristic disordered state (Fig. 1g). Upon addition of salt, as observed previously[30–32], a homogeneous solution of FUS-LC underwent phase separation as indicated by a rise in the turbidity of the solution (Supplementary Fig. 1a). Next, in order to directly visualize the droplet formation, we took advantage of the fact that FUS-LC does not contain any lysine residue and selectively labeled the N-terminal amine using the amine-reactive AlexaFluor488 succinimidyl ester (NHS ester) (Fig. 1c). This labeling strategy also allowed us to perform (selective) orthogonal dual labeling for FRET studies (see below). Using AlexaFluor488-NHS-labeled FUS-LC, we imaged the droplets using confocal microscopy (Fig. 1h). These droplets exhibited liquid-like behavior as evident by rapid and complete fluorescence recovery after photobleaching (FRAP) (Fig. 1i). These observations indicated that FUS-LC undergoes liquid phase condensation upon the addition of salt and are in agreement with previous reports[25,30,31]. Next, in order to elucidate the conformational changes associated with phase separation, we performed intramolecular FRET measurements both in monomeric dispersed and condensed phases.

The FRET donor (AlexaFluor488) was installed using the N-terminal NHS chemistry, whereas, the acceptor (thiol-active AlexaFluor594-maleimide), was covalently linked using thiol-maleimide chemistry at a Cys position of the single-Cys variants created along the FUS-LC polypeptide chain (Fig. 1b, c). We created three single-Cys mutants of FUS-LC (Cys residues at residue positions 86, 108, and 148) that encompassed the significant part of the polypeptide chain from the N- to the C-terminus and allowed us to record three intramolecular distances from the N-terminal end (N-to-86, N-to108, and N-to-148). The orthogonal labeling chemistries (NHS labeling at N-terminal amine and thiol-maleimide chemistry at Cys residues) yielded three FRET constructs selectively labeled with a donor (AlexaFluor488) and an acceptor (AlexaFluor594) (Fig. 1c). As a prelude to performing more

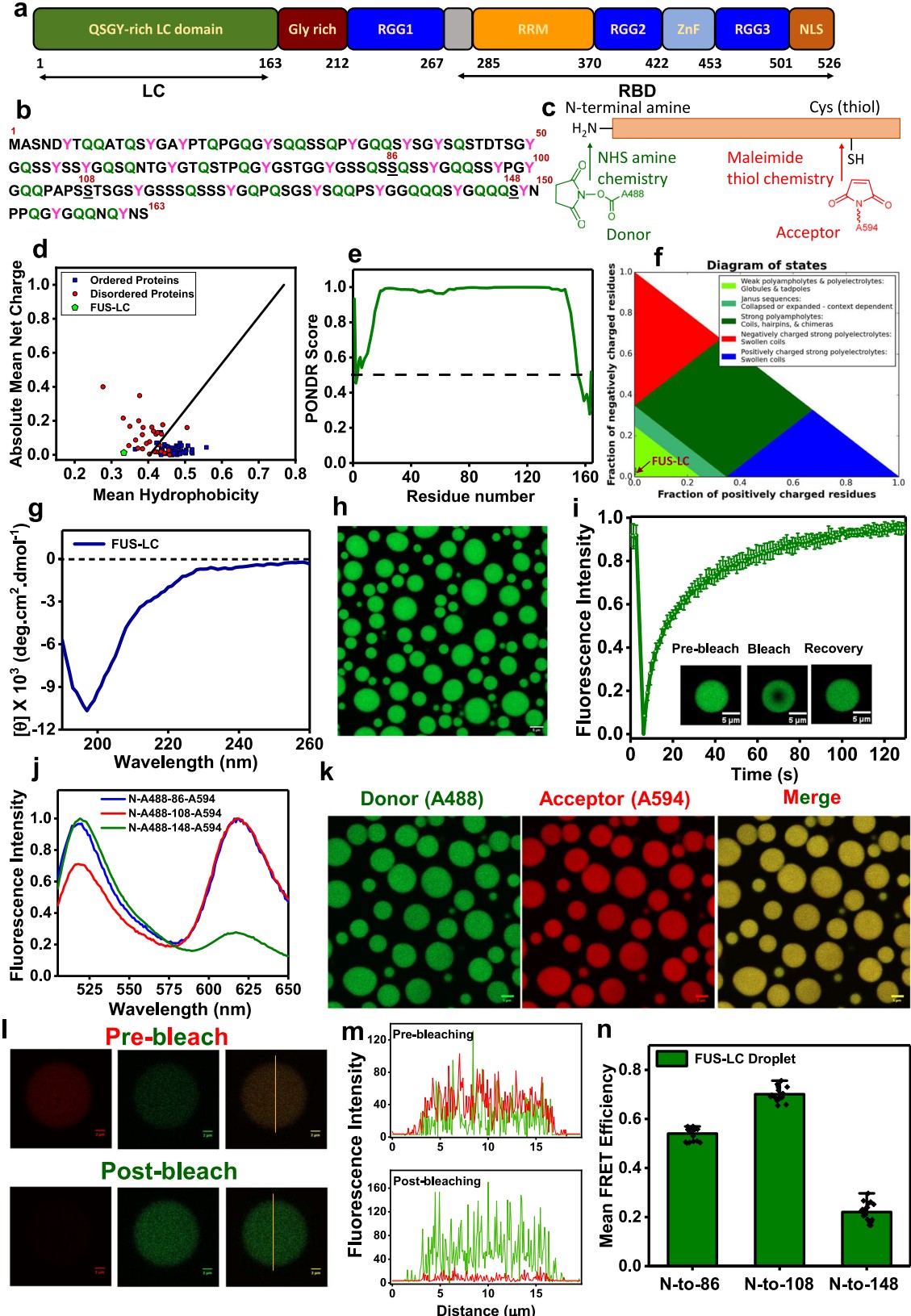

advanced single-molecule FRET experiments, we carried our ensemble steady-state FRET measurements both in spectroscopy and microscopy formats. The dual-labeled FUS-LC constructs exhibited energy transfer both in the monomeric dispersed state (Fig. 1j) and in the droplet phase as evident by an overlapping two-color confocal microscopy image (Fig. 1k). We next performed single-droplet

acceptor photobleaching experiments in a droplet-by-droplet manner. To determine the residue length-dependent FRET efficiency within these condensates, the acceptor was photobleached, and a subsequent increase in donor intensity was recorded (Fig. 1l, m), which was used to extract ensemble FRET efficiency within individual condensates (Fig. 1n). The FRET efficiency for the N-to-108 construct was

**Fig. 1 | Sequence architecture and characterization of FUS-LC liquid condensates. a** Domain architecture of full-length FUS. **b** The amino acid sequence of FUS-LC highlighting the tyrosine and glutamine residues. Residue positions for single-cysteine mutants are underlined. **c** Schematic of orthogonal labeling chemistry utilized for site-specific labeling of N-terminal with NHS ester of Alexa-Fluor488 and cysteine with thiol-active AlexaFluor594-maleimide. **d** The mean hydrophobicity vs. the mean net charge for a range of natively ordered and disordered proteins, FUS-LC is represented in green. **e** Predictor of Natural Disordered Regions (PONDR) showing unstructured FUS-LC. **f** A sequence annotated diagram of IDP states based on the fraction of negative and positive charged residues shows FUS-LC as compact globules and tadpole-like conformations. **g** A far-UV CD spectrum of FUS-LC indicating random-coil conformation. **h** Confocal image of FUS-LC droplets containing 0.1% AlexaFluor488-labeled protein (scale bar 5 μm). **i** FRAP kinetics of multiple droplets (*n* = 12) represented by the mean and standard

deviation (0.1% AlexaFluor488-labeled FUS-LC was used for FRAP measurements). **j** Ensemble steady-state fluorescence emission spectra of dual-labeled FUS-LC showing donor and acceptor fluorescence upon donor excitation at 488 nm. **k** Two-color Airyscan confocal image of FUS-LC droplets formed in the presence of 0.05% FUS-LC labeled with AlexaFluor488 at the N-terminal and AlexaFluor594 at Cys 86 (N-A488-86-A594) (scale bar 5 μm). **l** A representative image of acceptor photobleaching FRET of individual dual-labeled FUS-LC condensates (scale bar 2 μm) and **m** fluorescence intensity profiles showing a decrease in the acceptor fluorescence intensity and an increase in the donor fluorescence intensity upon acceptor photobleaching. The imaging was independently repeated 5 times with similar observations. **n** Mean FRET efficiencies in FUS-LC condensates were estimated from the acceptor photobleaching for three constructs varying the intramolecular distance, namely, N-to-86, N-to-108, and N-to-148. Data represent mean ± SD for *n* = 9 droplets.

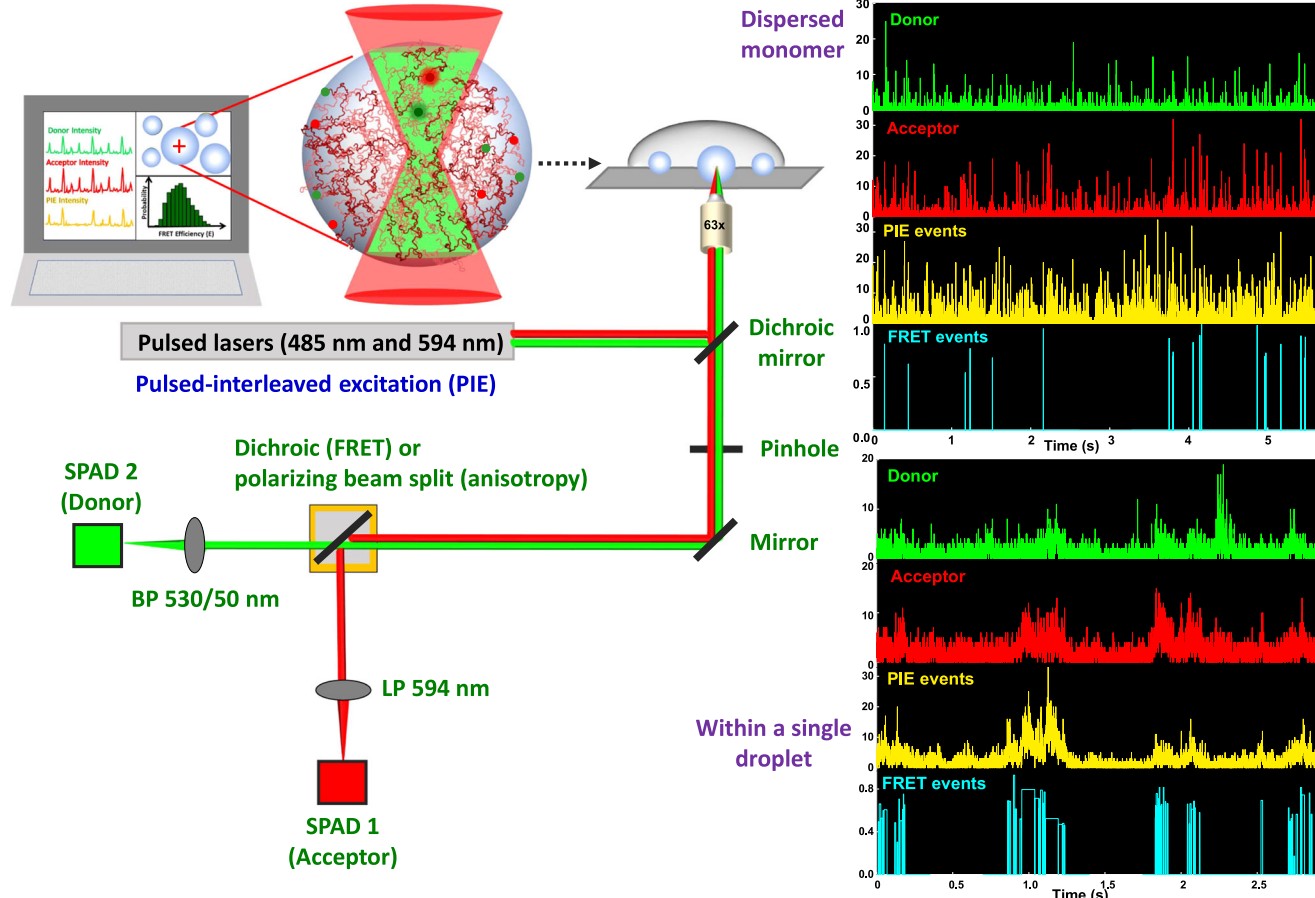

**Fig. 2 | Experimental design for single-droplet single-molecule measurements.** The schematic of our single-molecule microscopy setup (MicroTime 200, Pico-Quant). The major components are an excitation system with two picosecond pulsed lasers (485 nm and 594 nm), an inverted microscope, and the confocal detection system consisting of an integrated system of dichroic mirrors, pinhole, bandpass filters, and single-photon avalanche diodes (SPADs) as detectors. Representative time traces displaying fluorescence bursts in donor and acceptor channels and corresponding PIE and FRET events recorded in the monomeric dispersed (top) and droplet (bottom) phases are also shown. See "Methods" for more details.

significantly higher than the N-to-86 and N-to-148 constructs. A higher FRET efficiency for the N-to-108 construct hinted at the presence of some long-range interactions in the polypeptide chain within these condensates. These ensemble FRET experiments are not capable of discerning conformational distribution and dynamics but provide the groundwork for carrying out more advanced single-molecule FRET measurements. Therefore, to detect and characterize the co-existing conformationally distinct subpopulations and their interconversion, we next set out to perform our single-molecule FRET measurements both in the dispersed and condensed phases.

## Experimental design for single-droplet single-molecule FRET
In this section, we provide a brief description of the experimental design for carrying out single-molecule experiments within individual condensates. A more detailed description of the setup, experiments, data acquisition, and data analysis are provided in Supplementary Information. Our single-molecule FRET experiments were performed using the two-color pulsed-interleaved excitation (PIE) mode on a time-resolved confocal microscope (Fig. 2). The PIE-FRET methodology permitted us to identify the low FRET states by eliminating the contribution of the zero-FRET efficiency peak which arises due to the

acceptor photobleaching during the transit time[50]. In the case of the dispersed solutions, the dual-labeled protein concentration of 75–150 pM was sufficient to obtain single-molecule fluorescence bursts that arise due to freely diffusing fluorescently labeled protein molecules within the femtoliter confocal volume. Photon bursts separated into the donor and acceptor channels provide the footprints of individual molecules that diffuse in and out of the confocal volume and allow us to estimate the FRET efficiencies of individual diffusion events. In order to perform such single-molecule FRET experiments on individual condensates, a much lower concentration of the dual-labeled protein (5–10 pM) was needed in the condensed phase. Phase separation was induced from a mixture of unlabeled and dual-labeled FUS-LC so that the dual-labeled protein concentration was ~5–10 pM and the total protein concentration was 200 μM. We placed the solution on a coverslip surface and allowed it to settle for ~5 min after which most droplets got immobilized onto the glass surface. We then chose large immobilized droplets (3–7 μm diameter) that are much larger than the focal spot and focused the lasers inside these droplets (~2 μm from the surface into the droplet). We used this procedure for single-droplet single-molecule FRET, FCS, and fluorescence anisotropy measurements. Figure 2 shows representative time traces containing bursts in the dispersed phase and within single droplets. Using the PIE, only the bursts originating from the dual-labeled molecules qualify as FRET events for constructing FRET-efficiency histograms. The bursts originating from the droplets have longer duration compared to the monomeric protein which is attributed to the densely crowded environment and a slower diffusion within the condensed phase leading to a much larger number of excitation-emission cycles of the fluorophores during the transit time through the confocal volume. In the next two sections, we describe our single-molecule FRET results obtained in the monomeric and condensed phases of FUS-LC.

### Single-molecule FRET reveals two coexisting structural subpopulations of FUS-LC in the monomeric form

We performed our single-molecule FRET experiments with three dual-labeled constructs of FUS-LC varying the intramolecular distance (N-to-86, N-to-108, and N-to-148). Under the solution condition (pH 7.4, 250 mM NaCl), FUS-LC remains monomeric up to a concentration of 50 μM as evident by our dynamic light scattering measurements (Supplementary Fig. 1b, c). We performed single-molecule FRET experiments using 75–150 pM of dual-labeled FUS-LC. The single-molecule PIE-FRET efficiency histogram for the N-to-86 construct exhibited a unimodal distribution with a peak at ~0.76 (Fig. 3a) that corresponds to a mean inter-dye distance of ~45 Å. The observed mean FRET efficiency was significantly higher than the calculated mean FRET efficiency expected for a chain in a good solvent (Supplementary Table 2) suggesting that the FUS-LC chain is considerably collapsed. Next, we chose two other constructs (N-to-108 and N-to-148) having larger intramolecular distances and expected much lower FRET efficiency. In contrast to our expectation, we observed an unusual bimodal distribution in the single-molecule FRET efficiency histogram for both constructs. The construct having a 108-residue separation (N-to-108) exhibited two FRET efficiency peaks at ~0.97 and ~0.80 corresponding to distances of ~30 Å and ~43 Å, respectively. Whereas the N-to-148 construct showed peaks at ~0.73 and ~0.10 (Fig. 3b, c) corresponding to distances of ~46 Å and ~79 Å, respectively. These FRET efficiency histograms capture essential structural features indicating the presence of at least two predominant structural subpopulations in the monomeric conformational ensemble (Fig. 3d). These distinct subpopulations could involve compact S-shaped/paperclip-like (high-FRET) and partially extended tadpole-like (low-FRET) conformers that are in equilibrium having an interconversion exchange rate much slower than the observation time (0.5 ms). A binning time of 1 ms did not significantly alter the histograms suggesting the conformational exchange between these structural

subpopulations could even be slower than 1 ms (Supplementary Fig. 2a, b). Such structural distributions can satisfactorily explain the observed chain length-dependent inter-residue FRET efficiency histograms. The major population comprises the high-FRET S-shaped/paperclip-like conformers that can potentially arise due to the strong intrachain interactions driven by π–π interactions between multiple tyrosine residues and hydrogen bonding between glutamine side-chains (Fig. 1b). It is interesting to note that such compaction is observed in the N-terminal half of the polypeptide chain, whereas, the C-terminal segment adopts both compact and extended conformations possibly due to the presence of a larger number of proline residues. Taken together, our single-molecule FRET studies indicated that intrinsically disordered FUS-LC adopts two structurally distinct subpopulations having a varied extent of intrachain interactions. Next, we asked how conformational shapeshifting allows these intrachain interactions to turn into interchain interactions to promote phase separation of FUS-LC into liquid-like droplets.

### Single-droplet single-molecule FRET reveals a structural expansion and an increase in the conformational heterogeneity upon phase separation

We performed single-droplet single-molecule FRET measurements using all of the dual-labeled FUS-LC constructs (N-to-86, N-to-108, and N-to-148). The N-to-86 construct in the droplet phase also exhibited a unimodal FRET distribution with a lower mean FRET efficiency (~0.64) with a mean inter-dye distance of ~48 Å (Fig. 3e). The FRET histogram in droplets is associated with a broader distribution compared to the dispersed monomeric form that showed a mean FRET efficiency of ~0.76. This observation revealed the unwinding of the polypeptide chain that presumably allows the chains to participate in intermolecular interactions driving phase separation. The other two FRET constructs (N-to-108 and N-to-148) that showed bimodal FRET distribution in the monomeric form also exhibited a broadened distribution and a decrease in the energy transfer efficiencies upon phase separation (Fig. 3f, g). In the case of the N-to-108 construct, the mean FRET efficiencies of the two populations remained largely unaltered compared to the monomeric dispersed phase; however, the contribution of the low-FRET states grew with a concomitant broadening of the distribution. These results revealed that the N-terminal end and residues near the 108th position are involved in long-range contacts that are persistent on the millisecond timescale. For the N-to-148 construct, the low-FRET states with a mean FRET efficiency of ~0.30 constitute a major subpopulation within the droplet phase. It is interesting to note that these low-FRET states coexist with the high-FRET states in the condensed phase indicating conformational shapeshifting within the droplets occurs on a much slower timescale (≫1 ms) than the typical observation time (Supplementary Fig. 2c, d). A shift in the FRET efficiency towards lower values accompanied by a growth in the lower efficiency population in the droplet phase signifies that partially extended conformers constitute the major subpopulation in the conformational ensemble along with the presence of compact paperclip-like conformers within the condensed phase. These findings revealed that the compact FUS-LC conformational ensemble undergoes considerable unwinding engendering more structural plasticity and heterogeneity that allow the dissolution of intramolecular interactions and the formation of new intermolecular contacts promoting phase separation (Fig. 3h). Glutamine and tyrosine residues in the condensed phase can participate in a dynamic network of intermolecular hydrogen bonding and π–π interactions within droplets giving rise to a highly condensed network fluid. These dynamic interactions can undergo making and breaking on a characteristic timescale giving rise to the internal viscoelastic behavior of these condensates. Therefore, next, we set out to study the polypeptide chain diffusion and dynamics within individual condensates on a wide range of timescales.

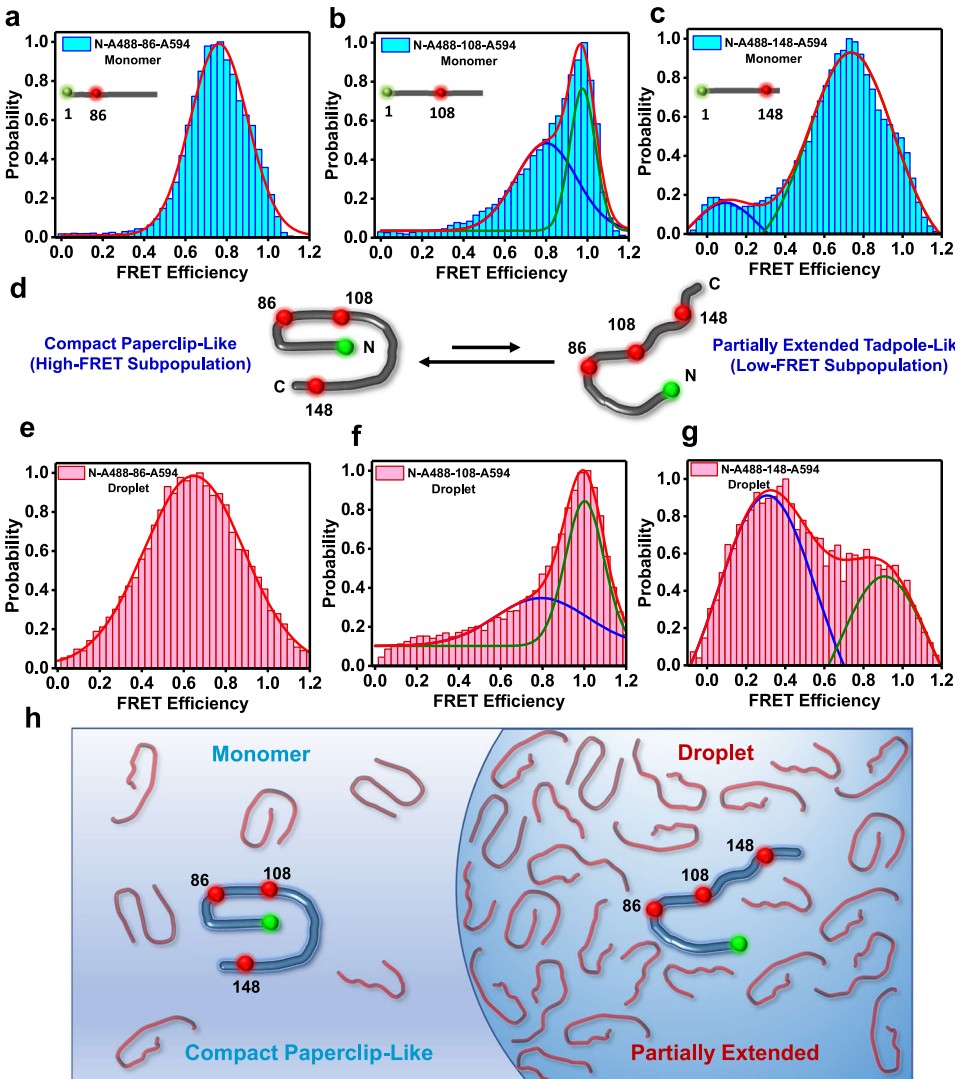

**Fig. 3 | Single-molecule FRET in monomeric dispersed and droplet phases of FUS-LC. a–c** Single-molecule FRET histogram of monomeric FUS-LC in the monomeric dispersed phase for dual-labeled **a** N-to-86, **b** N-to-108, and **c** N-to-148 constructs. The total number of events was > 20,000 and the number of events at maxima was 4217 (**a**), 1995 (**b**), and 2990 (**c**). **d** A schematic showing the coexistence of compact paperclip-like and extended tadpole-like conformers. **e–g** Single-droplet single-molecule FRET histograms for FUS-LC in condensed phase for dual-labeled **e** N-to-86, **f** N-to-108, and **g** N-to-148 constructs. The total number of events

was >20,000 and the number of events at maxima was 1329 (**e**), 2711 (**f**), and 831 (**g**). The FRET efficiency values and estimated inter-dye distances are shown in Supplementary Tables 2 and 3. The binning time was 0.5 ms. A binning time of 1 ms also yielded similar FRET histograms (Supplementary Fig. 2). The Förster radius of the FRET pair (Alexa488-Alexa594) used was 54 Å. See "Methods" for more details of experiments, data acquisition, data analysis, and distance estimation. **h** A schematic depicting the structural unwinding of compact conformers into partially extended conformers upon phase separation.

## Translational and rotational dynamics within individual condensates reveal the formation of a viscoelastic network fluid

In order to investigate the chain diffusion within individual condensates, we performed single-droplet FCS measurements using AlexaFluor488-labeled FUS-LC. FUS-LC exhibited a diffusion time of ~0.13 ms in the monomeric dispersed form. The translational diffusion time increased 400 times to ~50 ms in the droplet indicating a highly crowded and viscous environment within the droplets presumably due to the formation of a dynamic network of intermolecular interactions (Fig. 4a, b). We envisaged such a network of interactions giving rise to the condensate formation would impede the reorientation dynamics of polypeptide chains involved in intermolecular multivalent interactions. In order to delineate the role of reorientation dynamics, we employed site-specific single-droplet fluorescence (polarization) anisotropy measurements that report the extent of rotational flexibility of the polypeptide chain. For fluorescence anisotropy experiments, we chose four sites along the polypeptide chain and labeled single-Cys

variants of FUS-LC (A16C, S86C, S108C, and S148C) using thiol-active fluorescein-5-maleimide that contains a shorter linker than Alexa dyes, and therefore, can report the rotational flexibility of the polypeptide chain without exhibiting a significant local depolarization. All residue locations exhibited a sharp increase in the steady-state fluorescence anisotropy in the condensed phase compared to the dispersed monomeric phase (Fig. 4c, d). These results indicated dampening of the rotational flexibility of the FUS-LC chain within droplets. Notably, the 108th position exhibited a higher anisotropy value in both the monomer and droplet phases suggesting the possibility of some persistent long-range contacts that are in accordance with our single-molecule FRET data. Although steady-state fluorescence anisotropy measurements suggested reorientation restraints within condensates, these measurements do not allow us to discern the contributions of distinct modes of rotational dynamics. Next, we employed single-droplet picosecond time-resolved fluorescence anisotropy measurements that can discern the various modes of rotational dynamics.

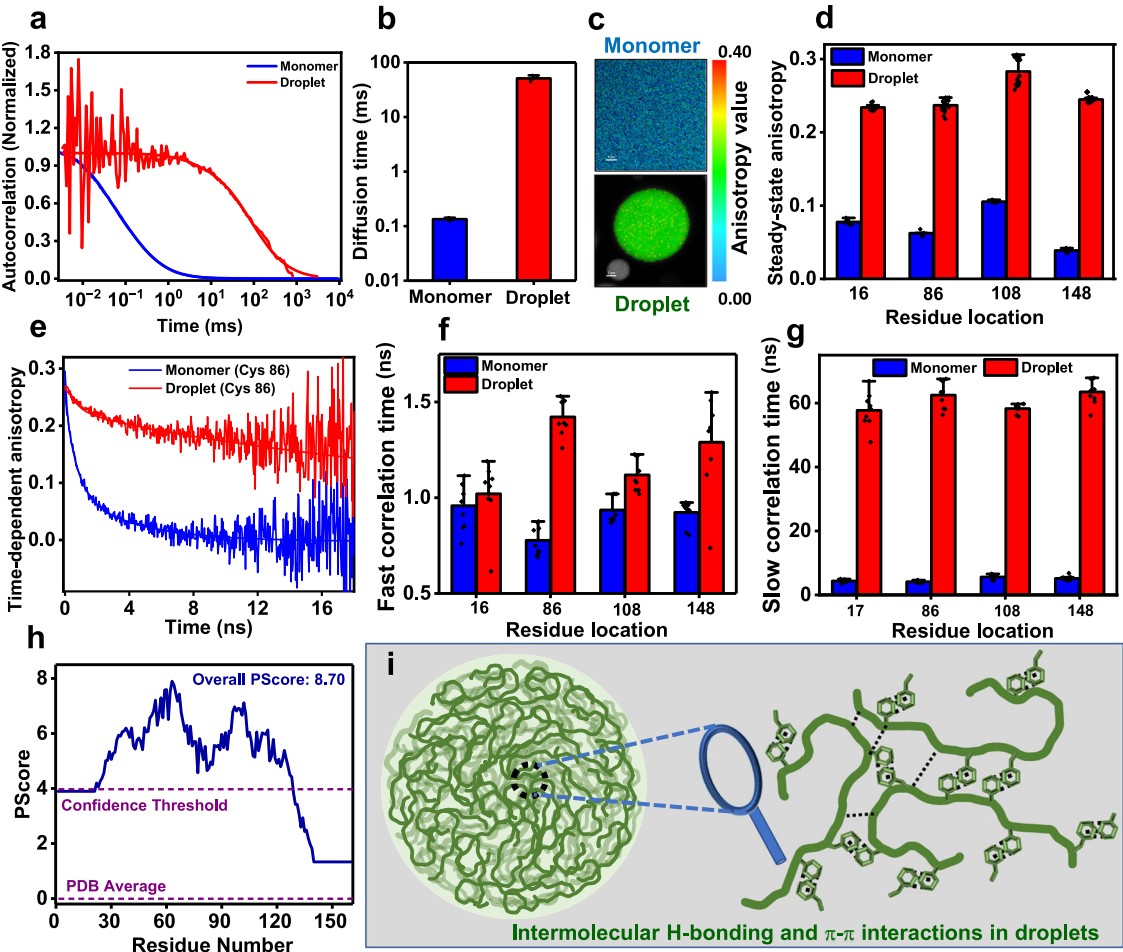

**Fig. 4 | Translational and rotational dynamics. a** Autocorrelation plots (normalized) acquired from FCS measurements performed for AlexaFluor488 labeled FUS-LC in monomer and within single droplets (unnormalized FCS autocorrelation plots are shown in Supplementary Fig. 3a, b). **b** The diffusion time of FUS-LC in monomeric and condensed phases estimated from FCS. Data represent mean ± SD for $n = 5$ independent samples. FCS measurements were performed in the presence of 10 nM (in dispersed monomer) and 1–3 nM (in droplet) of AlexaFluor488-labeled FUS-LC. **c** Representative fluorescence anisotropy images for a dispersed phase (scale bar 5 μm) and a single droplet (scale bar 2 μm) showing anisotropy heatmap. **d** Steady-state fluorescence anisotropy values for FUS-LC in monomeric and within individual condensates. Fluorescence anisotropy was measured using fluorescein-5-maleimide-labeled FUS-LC at residue positions 16, 86, 108, and 148. Data represent mean ± SD for $n = 6, 5, 8, 8$ independent samples at positions 16, 86, 108, and 148 respectively, for monomers and $n = 24$ for droplets. The fluorescence lifetime did not exhibit a significant change from monomeric dispersed to condensed phase. **e** Representative picosecond time-resolved fluorescence anisotropy decay profiles for FUS-LC labeled at residue position 86 in monomer and droplets. Solid lines are fits obtained from biexponential decay analysis. Time-resolved anisotropy decays for the other three locations (16, 108, and 148) are shown in Supplementary Fig. 3c–e. **f** Fast rotational correlation times and **g** slow rotational correlation times for residue locations 16, 86, 108, and 148 recovered from decay analyses. Data represent mean ± SD ($n = 9$ independent samples). Rotational correlation times are included in Supplementary Table 4. **h** Predictor of phase separation of IDPs based on the propensity to form long-range planar π–π contacts calculated as Pscore value for FUS-LC. **i** A schematic representing a dense network of intermolecular π–π contacts and hydrogen bonding within FUS-LC condensates.

In order to temporally resolve the distinct molecular events, we utilized the highly sensitive picosecond time-resolved fluorescence anisotropy decay measurements that permit us to probe the depolarization kinetics of fluorescence anisotropy from the time-zero anisotropy value due to various modes of rotational relaxation[51,52]. In the case of monomeric IDPs, the depolarization kinetics follow a typical multiexponential decay function. Such anisotropy decay functions typically involve a fast rotational correlation time representing the local wobbling-in-cone motion of the fluorophore and slow rotational correlation times corresponding to backbone dihedral angle fluctuations and long-range reorientation dynamics[53–55]. As expected, the anisotropy decay profiles for monomeric FUS-LC recorded at different residue locations were satisfactorily described by a biexponential decay model giving rise to two well-separated correlation times: fast rotational correlation time representing a local probe motion (~1 ns) and a slow rotational correlation time (~4–5 ns) corresponding to dihedral and long-range reorientation motions (Fig. 4e–g). We then

measured fluorescence anisotropy decay kinetics within individual droplets and observed that the depolarization kinetics slowed down considerably. The fast local correlation time exhibited a slight increase, whereas, the slow rotational correlation displayed a sharp increase from ~5 ns to ~60 ns at all residue locations (Fig. 4e–g). Such an increase in the slow correlation time within condensates indicated a dampening of the chain reorientation dynamics presumably due to the formation of a network via interchain physical crosslinks. Based on the previous studies on FUS-LC indicating the role of the glutamine and tyrosine residues in hydrogen bonding, π–sp², and hydrophobic interactions[25], we postulate that these intermolecular contacts form a dense network of physical crosslinks within the FUS-LC condensates (Fig. 1b). The PScore analysis[18] that quantifies the π–π contacts in proteins revealed a high propensity of π–π interactions mediated phase separation of FUS-LC which contains 24 tyrosine and 37 glutamine residues (Fig. 4h and Supplementary Fig. 3f). Taken together, our results on chain dynamics coupled with structural subpopulations

indicate the presence of a network of intermolecular interactions as depicted in our schematic (Fig. 4i). Such a network of dynamic physical crosslinks can slow down translational and rotational diffusion ensuing a viscoelastic network fluid within the condensates. Next, we asked if our unique structural and dynamical readouts of wild-type FUS-LC can detect and distinguish the altered phase behavior of a disease-associated mutation.

## A disease-associated mutant alters the phase behavior by modifying conformational distribution and dynamics

Several mutations in the disordered LC domain, RNA-binding domain, and NLS are associated with various neurodegenerative diseases. One such mutation in the LC domain (G156E) is a patient-derived, clinically relevant mutation that functions by modulating the phase behavior and aggregation propensity of FUS[34,56,57]. Thus, we next set out to study the effect of this disease-related mutation on the conformational characteristics and phase behavior of FUS-LC. We created a single-point mutant (G156E) and recombinantly expressed this construct. The CD spectrum of G156E FUS-LC indicated a disordered conformation and showed no significant changes in the secondary structural contents as compared to the wild-type FUS-LC (Supplementary Fig. 4a). Next, we began with the phase separation assay of G156E FUS-LC. A rise in turbidity values of the protein solution in the presence of salt (Supplementary Fig. 4b) indicated the formation of droplets capable of recruiting dual-labeled wild-type FUS-LC as confirmed by our two-color confocal fluorescence imaging (Fig. 5a). To predict the effect of this mutation on the phase behavior of FUS-LC, we employed a bioinformatics analysis tool namely catGranule[58], which computes the phase separation propensity of proteins based on their FG and RG contents, disorder, and RNA-binding propensity. As predicted by catGranule[58] (Supplementary Fig. 4c), the phase separation propensity of G156E FUS-LC was slightly lower in comparison to the wild-type FUS-LC as indicated by relatively lower turbidity (Supplementary Figs. 1a and 4b) and a higher saturation concentration compared to wild-type FUS-LC (Fig. 5b). Unlike for the low-complexity domain of FUS, the phase separation propensity of full-length FUS appears to remain unaffected by the G156E mutation[56,57]. However, this mutation accelerates aggregation of full-length FUS and can potentially change the physical properties of condensates. To further characterize these G156E FUS-LC droplets, we performed FRAP measurements which showed a slower and lower fluorescence recovery compared to wild-type FUS-LC (Fig. 5c). In order to get insights into the polypeptide chain diffusion within individual droplets, we performed single-droplet FCS measurements which indicated a slightly slower translational diffusion (-70 ms) in comparison to wild-type FUS-LC droplets (-50 ms) (Fig. 5d and Supplementary Fig. 4d). These results suggested that the G156E mutation alters the phase behavior and the condensed phase exhibits more restrained diffusion.

Next, in order to gain structural insights into the wild-type FUS-LC chain within G156E FUS-LC condensates, we performed ensemble single-droplet FRET using acceptor photobleaching in the confocal fluorescence microscopy format. Ensemble single-droplet FRET indicated the retention of long-range contacts between the N-terminal and residue location 108 similar to the wild-type FUS-LC droplets (Fig. 5e). To obtain insights into the structural subpopulations and conformational dynamics, we then performed single-droplet single-molecule FRET measurements (Fig. 5f–h). The wild-type FRET-pair-labeled LC acts as a reporter of the environment, the extent of interactions, and conformational distributions of the densely packed surrounding polypeptide chains within condensates of mutant FUS-LC (G156E). FRET efficiency histograms for the N-to-86 and N-to-108 constructs within G156E droplets were similar to those observed for wild-type FUS-LC droplets. Interestingly, the FRET efficiency histogram for the N-to-148 construct exhibited a slight shift towards lower FRET efficiency compared to the wild-type FUS-LC droplets (Fig. 5i and Supplementary

Table 3). This reduction in FRET efficiency of both populations indicated more expansion of the C-terminal segment of the chain that facilitates the formation of a network of intermolecular contacts within the G156E FUS-LC condensates. Such a conformational unwinding in the mutant, presumably due to the presence of an additional negative charge, can offer a larger number of multivalent interactions leading to a more densely packed interior of the condensate and more dampened translational and rotational dynamics. This observation is consistent with a slight increase in the steady-state fluorescence anisotropy within G156E FUS-LC condensates (Supplementary Fig. 4e) and a slower internal diffusion corroborating our FRAP and FCS results. Taken together, our results revealed an unraveling of the polypeptide chain within condensates of the G156E mutant promoting increased interchain interactions and a dense network which can further facilitate pathological aggregation of the mutant FUS-LC as previously observed[34,56,57]. Next, we wanted to distinguish the secondary structural features that govern the condensate properties of wild-type and mutant FUS-LC.

## Single-droplet vibrational Raman spectroscopy supports the altered phase behavior of the disease-associated mutant

In order to probe the secondary structural features of FUS-LC, we performed single-droplet vibrational Raman spectroscopy that allowed us to obtain insights into polypeptide structure and organization within individual droplets[59,60]. Focusing a laser beam into a droplet permits us to obtain a Raman spectrum that contains the characteristic vibrational signatures corresponding to the polypeptide backbone (amide I and amide III) and side chains (aromatic and aliphatic residues). These vibrational signatures provide us with vital information on the secondary structural distributions and intermolecular interactions within a single condensate (Fig. 5j). Amide I vibrational band (1630–1700 cm$^{-1}$) originates primarily due to the C=O stretching vibrations of the polypeptide backbone, while the amide III band (1230–1320 cm$^{-1}$) involves a combination of C-N stretch and N-H bending motions of the backbone. Together these amide bands constitute the secondary structural marker bands highlighting the secondary structural elements present in the proteins[25,59–62]. We observed a broad amide I band for wild-type FUS-LC droplets indicating the presence of disordered conformations and considerable conformational heterogeneity in the condensed phase (Fig. 5k) corroborating our single-molecule FRET results. Deconvolution of the baseline-corrected amide I indicated the presence of two major peaks representing extended/unordered (-1692 cm$^{-1}$) and some ordered structural elements in the protein-rich dense phase. The peak at -1671 cm$^{-1}$ corresponds to ordered structural elements that might contain β-sheet with some minor α-helical contents (-1652 cm$^{-1}$) and can represent the signature of compact S-shaped/paperclip-like states as observed in our FRET experiments (Fig. 5l). On the contrary, the deconvolution analysis of the amide peak for G156E FUS-LC droplets revealed a largely unstructured state (-1666 cm$^{-1}$ and -1692 cm$^{-1}$) within droplets corroborating our FRET data (Fig. 5m). Nevertheless, a small increase in the full width at half maximum (FWHM) of the amide I band from wild-type (49.6 ± 0.6 cm$^{-1}$) to mutant droplets (51.4 ± 0.6 cm$^{-1}$) could further support more conformational heterogeneity in the case of the G156E mutant. More structural unzipping and heterogeneity in the case of the mutant allows a much denser network of interaction as observed in our single-molecule FRET, FCS, and anisotropy measurements. This is also supported by the Raman intensity ratio of the tyrosine Fermi doublet ($I_{850}/I_{830}$), which indicates the hydrogen bonding propensity of the phenolic hydroxyl group of tyrosine with surrounding water molecules or in other words, determines the average solvent accessibility of tyrosine residues. This ratio exhibited a small but measurable change from -1.4 for wild-type droplets to -1.6 for mutant droplets suggesting a slightly higher solvent exposure of tyrosine residues within G156E FUS-LC droplets resulting

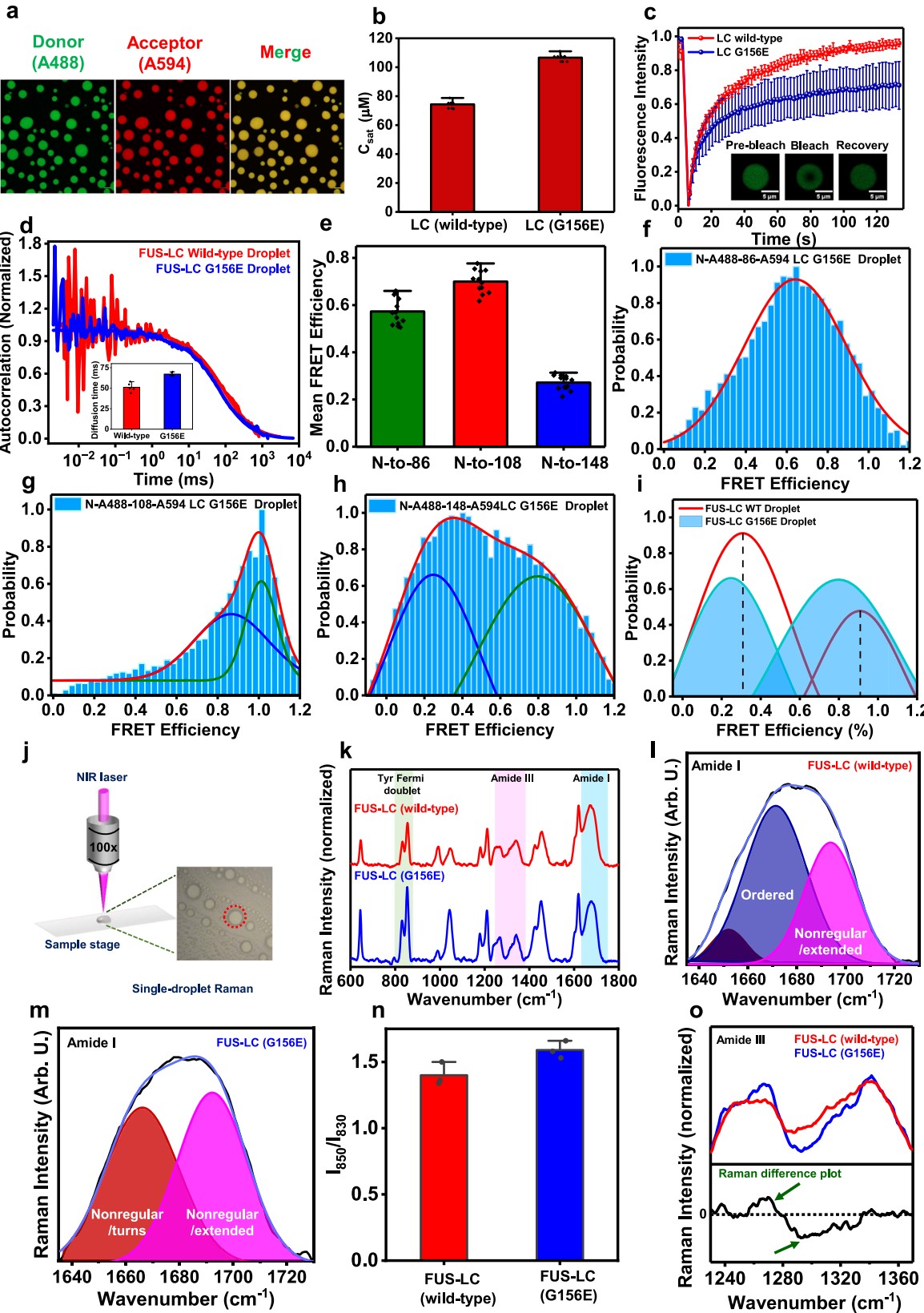

from relatively expanded conformers facilitating a larger extent of intermolecular π-π contacts between more expanded chains within G156E FUS-LC droplets (Fig. 5n). We further zoomed into the amide III region for wild-type and G156E droplets and constructed a Raman difference plot for the comparison. A positive band centered at ~1270 cm⁻¹ (nonregular/turns) and a negative band at ~1300 cm⁻¹

highlighted a higher content of disordered conformations within mutant droplets as compared to wild-type droplets (Fig. 5o). Our single-droplet vibrational Raman studies capture the key structural differences between wild-type and mutant FUS-LC droplets and are in agreement our single-molecule FRET, FCS, FRAP, and anisotropy results. Taken together, the disease-associated mutant (G156E) of

**Fig. 5 | The effect of disease-associated mutation (G156E) on FUS-LC. a** Airyscan confocal imaging showing G156E FUS-LC droplets recruit wild-type dual-labeled FUS-LC (scale bar 5 µm). **b** Saturation concentrations ($C_{sat}$) of wild-type and G156E FUS-LC were estimated using centrifugation. Data represent mean ± SD ($n = 9$ independent reactions). **c** FRAP kinetics of AlexaFluor488-labeled FUS-LC within wild-type (same as in Fig. 1i) and G156E mutant droplets. Data represent mean ± SD ($n = 8$ droplets). **d** Normalized FCS autocorrelation plots with mean diffusion times (Inset) for wild-type (same as in Fig. 4a, b) and G156E droplets. Data represent mean ± SD ($n = 5$ droplets). **e** Mean FRET efficiency obtained from acceptor photobleaching of G156E FUS-LC droplets. Data represent mean ± SD ($n = 13$ droplets). Single-droplet single-molecule FRET histograms in G156E FUS-LC droplets using dual-labeled **f** N-to-86, **g** N-to-108, and **h** N-to-148 constructs. The total number of events was >20,000 and the number of events at maxima was 1019 (**f**), 1725 (**g**), and 698 (**h**). **i** Overlay of fitted subpopulations for wild-type and G156E FUS-LC droplets

(shaded) for the N-to-148 construct showing a shift towards a lower FRET efficiency within G156E FUS-LC droplets. **j** A schematic for our single-droplet Raman measurements by focusing a near-infrared (NIR) laser into a single droplet of FUS-LC. **k** Mean Raman spectra for wild-type and G156E FUS-LC droplets. ($n = 3$ droplets, for individual spectra see Supplementary Fig. 5.) Raman spectra are normalized with respect to the amide I band at -1673 cm$^{-1}$. **l, m** Gaussian deconvolution of separately baseline-corrected amide I of wild-type FUS-LC (**l**) and G156E FUS-LC (**m**) to estimate the composition of various secondary structural elements. The black and blue solid lines represent the actual data and the cumulative fit, respectively. The colored solid lines represent the Gaussian peaks obtained after deconvolution. **n** The ratio of the tyrosine Fermi doublet ($I_{850}/I_{830}$) estimated from the peak intensities. Data represent mean ± SD ($n = 3$ droplets). **o** The Raman difference plot of the amide III bands of FUS-LC and G156E FUS-LC droplets (arrow indicates the difference of interest). See Supplementary Table 5 for peak assignments.

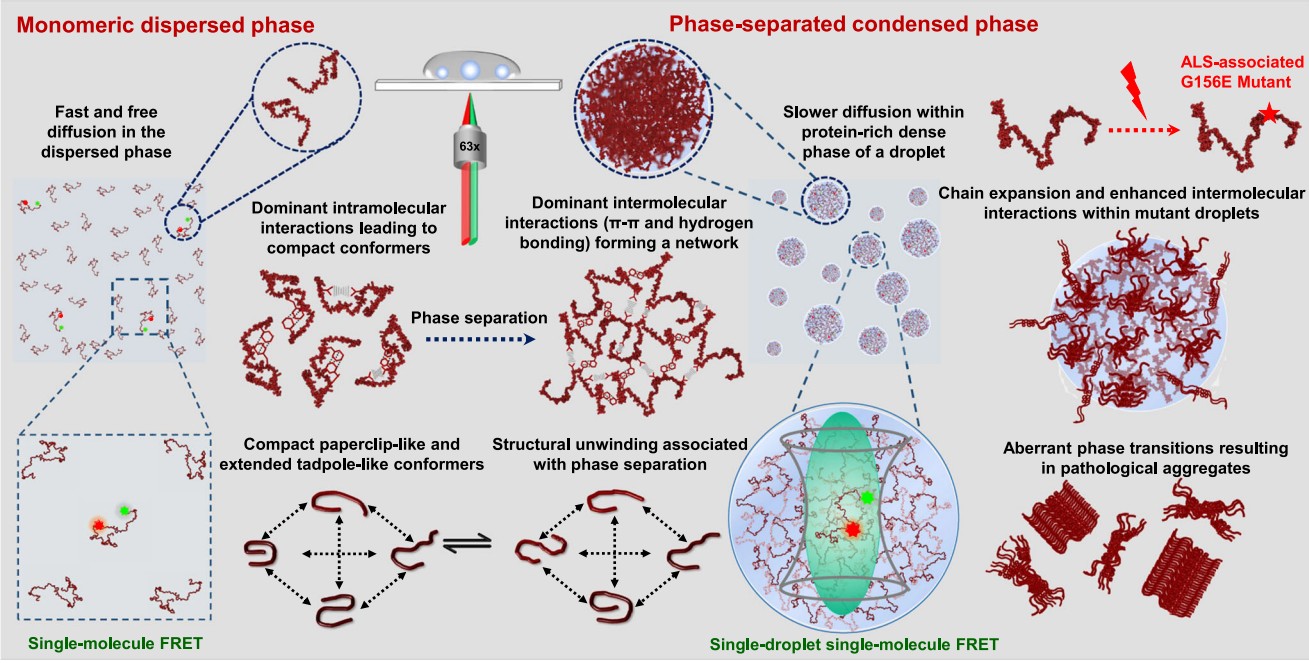

**Fig. 6 | A schematic of conformational shapeshifting during phase separation.** A summary of our single-molecule FRET, FCS, fluorescence anisotropy, Raman spectroscopy experiments, and the impact of a pathological mutation in the phase

behavior of FUS-LC. Disordered polypeptide chains generated using PyMOL (Schrödinger, LLC, New York) are shown for a schematic illustration.

FUS-LC exhibits higher disorder facilitating more intermolecular association with altered material properties that can potentially promote aberrant phase transitions into solid-like pathological aggregates compared to the wild-type form.

## Discussion

In this work, we showed that the polypeptide chain of FUS-LC undergoes conformational shapeshifting from the monomeric dispersed phase to the condensed phase. We summarized all our observations in a schematic illustration in Fig. 6. By employing single-molecule FRET, we characterized the conformational distribution and dynamics of FUS-LC, site-specifically and orthogonally labeled with donor and acceptor fluorophores. Such single-molecule studies carried out at ultralow concentrations (-100 pM) allow us to unambiguously interrogate and characterize the monomeric form of the protein in a molecule-by-molecule manner that is not generally achievable by most other conventional structural tools. Our single-molecule FRET studies for freely diffusing molecules revealed that intrinsically disordered FUS-LC monomer exists as a heterogeneous ensemble of structures comprising chiefly two distinct, well-resolved, structural subpopulations. These subpopulations include compact S-shaped/paperclip-like

conformers and partially extended tadpole-like conformers exchanging on a much slower timescale (≫1 ms) than the observation time. Such slower conformational exchanges between extended and compact states can further be studied using surface-immobilized single-molecule FRET. The compaction of the disordered FUS-LC chain we observed in this study can be ascribed to the presence of extensive intramolecular interactions arising due to π–π stacking interactions between multiple tyrosine residues and hydrogen bonding interactions between glutamine side chains, which have previously been shown to drive phase separation of FUS-LC[25]. We propose that a large number of proline and glycine residues promote partial expansion, especially at the C-terminal part of the polypeptide chain. Such making and breaking of intramolecular noncovalent contacts give rise to rapidly fluctuating, intrinsically disordered, heterogeneous conformational ensemble in the monomeric form of FUS-LC. This conformational equilibrium in the infinitesimally dilute condition changes quite dramatically when the protein concentration is raised beyond a threshold concentration also known as the saturation concentration. Above this concentration, intermolecular chain-chain interactions become more dominant than intramolecular contacts, and thus, unzipping of compact conformers becomes facile due to the more

favorable intermolecular multivalent interactions driving liquid phase condensation. In the case of full-length FUS, electrostatic and cation-π interactions between the arginine-rich C-terminal and tyrosine-rich N-terminal domains function as the major drivers of FUS phase separation[63]. On the contrary, extensive computational and mutagenesis studies on the N-terminal domain of FUS revealed the role of multiple glutamine and tyrosine residues in the formation of hydrogen bonding, hydrophobic, and $\pi$–$sp^2$ contacts within FUS-LC condensates[25]. The slow conformational exchange between the conformers allows more interaction lifetime yielding a network of intermolecular interactions in the condensed phase. Such dynamically controlled conformational gymnastics can play a pivotal role in driving protein phase transitions. Additionally, previous studies have identified persistent secondary cross-β core structures within gel-like states of FUS-LC[36], which appear to be absent in liquid droplets. Our observation is consistent with the previous computational and structural studies on FUS-LC highlighting the presence of a compact state in the monomeric form and the role of conformational disorder in phase separation[25,31,35,49].

Indeed, our single-droplet single-molecule FRET studies revealed a symphony of conformational shapeshifting events within the dense phase of individual condensates as depicted in Fig. 6. Some of these unique molecular features involving distinct subpopulations generally remain skewed in traditional bulk experiments and can only be directly observed using a single-molecule tool. Single-molecule FRET coupled with FCS and picosecond time-resolved fluorescence anisotropy measurements allowed us to directly decipher the conformational distribution and dynamics within the dense phase. The solvent quality within this protein-rich dense phase is likely to be better than that in the monomeric form of the protein dispersed in water which is a poor solvent for a polypeptide chain. A better solvent quality, and hence a higher Flory scaling exponent, in the dense phase is expected to favor the conformational expansion. Such expanded conformers can also turn intramolecular contacts into transient intermolecular contacts favoring the condensed phase. The presence of a subpopulation of compact states suggests that the conformational interconversion is possibly associated with the making and breaking of multivalent interactions giving rise to liquid-like behavior within droplets. An interesting hypothesis can be posited based on the radial gradient of the physicochemical properties with a complex interplay of intra-chain, inter-chain, and chain–solvent interactions in the condensed phase[64]. These droplets could possess a radial distribution of conformational subpopulations based on the spatial locations within the condensate. Previous simulation studies on intrinsically disordered prion-like low-complexity domains have indicated such conformational heterogeneity within the dense phase[64,65]. Recent studies revealed that a small-world percolated network comprising a varied density of physical crosslinks can give rise to distinct conformational states and molecular orientations varying spatial location from the center to the interface of the condensates[64]. Similar studies on the full-length FUS showed pathological aggregation via phase separation by onset at the condensate interface and investigated the material properties and multiphasic nature of condensates from the core to the interface[66,67]. The spatial resolution of the confocal microscopy-based format used in our single-molecule experiments is inadequate for dissecting the radial distribution of conformers. The FRET efficiency distribution displayed by our observed single-molecule FRET histograms could potentially comprise such conformational distributions across the droplet locations. We would also like to point out that although we observed much broader peaks in the FRET efficiency histograms in droplets, we are unable to comment on the exact distances and the width of individual peaks since there can be some contributions from the shot noise due to low photon counts and the dye orientation factor

that can potentially contribute to the widths of the distribution. Our single-droplet FCS and fluorescence anisotropy results revealed that both translational diffusion and rotational chain reorientation dynamics are considerably slowed down within condensates. These findings hinted at the formation of a network of physical crosslinks giving rise to viscoelastic network fluid. Our results on a disease-associated mutant (G156E) indicated a more conformational plasticity promoting a denser network of intermolecular contacts as observed by single-molecule FRET, FRAP, FCS, and anisotropy measurements. Additionally, single-droplet vibrational Raman spectroscopy corroborated our results on altered conformational distribution for the mutant. Raman signatures for both backbone and sidechain markers indicated slightly more extended conformers and greater participation in chain-chain association within condensates. Such interactions can potentially promote aberrant liquid-to-solid phase transitions and accelerated aggregation associated with the pathological hallmark of the G156E mutant of FUS (Fig. 6).

In summary, our single-molecule experiments directly unveiled an intriguing interplay of conformational heterogeneity, structural distribution, and dynamics that crucially governs the course of phase transitions of prion-like low-complexity domains. Such low-complexity domains are present in FUS and other FET-family proteins and are essential in mediating the homotypic and heterotypic interactions driving the assembly into liquid-like functional condensates and solid-like pathological aggregates[66–68]. Post-translational modifications and mutations can alter the interplay between intra- and intermolecular interactions shifting the conformational equilibria both in the dispersed and condensed phases. Such altered interactions can give rise to changes in the viscoelastic material properties of biomolecular condensates, promoting aberrant phase transitions associated with ALS and FTD. Single-molecule FRET in combination with mutagenesis can offer a potent approach to discern the impact of disease-associated mutations and post-translational modifications on the conformational preference and phase behavior of IDPs/IDRs. Our findings on the role of conformational excursion in phase separation can have much broader implications for a wide range of phase-separating proteins involved in physiology and disease. For instance, tau, an intrinsically disordered neuronal protein associated with Alzheimer's disease, exhibits conformational subpopulations comprising compact paperclip/S-shaped and expanded states[43,69–71]. Phase separation of tau studied at the single-molecule resolution also revealed an increased conformational heterogeneity. We suggest that the sequence-encoded structural unwinding coupled with a dynamic control can expose the multivalency of polypeptide chains that promote ephemeral interactions resulting in biomolecular condensate formation. Deriving general principles from single-molecule studies on a wide range of phase-separating proteins and artificial polypeptides can provide the key mechanistic underpinning of macromolecular phase separation and pave the way for novel synthetic biology applications. Additionally, the development of multi-color, multi-parameter, super-resolved, single-droplet single-molecule FRET can offer unprecedented spatiotemporal resolution in studying intracellular heterotypic, multicomponent, and multiphasic biomolecular condensates and for exploring unchartered territories of biological phase transitions.

## Methods

### Bioinformatics analyses

Various bioinformatics tools were used for the sequence characterization of FUS-LC. Classification of Intrinsically Disordered Ensemble Regions (CIDER) (https://pappulab.wustl.edu/CIDERinfo.html)[46] for visualization of charged and hydrophobic amino acids and Predictor of Natural Disordered Regions (PONDR) (http://www.pondr.com/)[45] for

disorder propensity prediction were used. To determine the phase separation propensity catGranule (http://www.tartaglialab.com/)[58] and PScore (http://abragam.med.utoronto.ca/-JFKlab/)[18] (based on π–π interaction propensity) were used. Disorder and phase separation propensity plots were generated using the Origin software.

## Construct details and site-directed mutagenesis
All single cysteine and disease mutants were created by site-directed mutagenesis using the recombinant MBP-His$_6$-FUS-LC WT (Addgene plasmid # 98653; https://www.addgene.org/98653/; RRID: Addgene_98653) cloned in pTHMT vector which was a kind gift from Nicolas L. Fawzi. The primer sets used for introducing these point mutations have been listed in Supplementary Table 1. All mutations were confirmed by sequencing.

## Recombinant protein expression and purification
Wild-type and all the variants of FUS-LC were transformed in *E. coli* BL21(DE3) std cells, overexpressed, and purified using affinity chromatography, followed by gel-filtration chromatography. Bacterial cultures were grown at 37 °C, 220 rpm, to an O.D.$_{600}$ of 0.8–1. Protein overexpression was induced by adding 1 mM isopropyl-β-thiogalactopyranoside (IPTG) and further growing cultures at 37 °C for 4–5 h. Bacterial cells were harvested by centrifugation at 4 °C, 3220 × *g* for 30 min, and stored at −80 °C for future use. Cell pellets were resuspended in lysis buffer (20 mM sodium phosphate, 300 mM NaCl, 10 mM imidazole, pH 7.4) and were lysed by probe sonication at 5% amplitude, 15 s ON, and 10 s OFF for 20 min. The lysate was centrifuged at 4 °C, 15,557 × *g* for 1 h to remove the cell debris, and the supernatant was loaded onto a Ni-NTA column. The column was washed, and the bound protein was eluted with elution buffer (20 mM sodium phosphate, 300 mM NaCl, 300 mM imidazole, pH 7.4).

The N-terminal MBP-His$_6$ tag was cleaved by adding recombinantly expressed and in-house purified TEV protease at a 1:40 molar ratio (TEV: protein), followed by incubation at 30 °C for 1.5 h. It was then subjected to overnight dialysis at room temperature. The cleaved protein was passed through the Ni-NTA column to separate the uncleaved species and TEV protease and flowthrough were collected and concentrated using a 10 kDa MWCO Amicon filter. Concentrated protein was further loaded on HiLoad 16/600 Superdex-G-200 (GE) column equilibrated with the SEC buffer (20 mM CAPS, 150 mM NaCl, pH 11). SEC elution fractions were run on an SDS-PAGE gel to determine fractions containing protein of interest. Pure protein fractions were pooled, concentrated, and buffer-exchanged into 20 mM CAPS, pH 11 buffer using a PD-10 column. Pure protein was concentrated using a 3 kDa MWCO Amicon filter, and concentration was estimated by measuring absorbance at 280 nm ($\varepsilon_{280} = 30,720$). Pure protein was flash-frozen and stored at −80 °C.

## Circular dichroism (CD) measurements
Far-UV CD spectra were recorded using a Chirascan spectrophotometer (Applied Photophysics, UK) in a quartz cuvette of 1 mm path length. Wild-type and G156E FUS-LC were diluted to 10 μM in 20 mM sodium phosphate buffer, pH 7.4. Measurements were made for buffer and monomeric FUS-LC. Recorded absorption spectra were averaged over 10 scans, followed by blank subtraction using ProData Viewer version 4.1.9 software, and plotted using the Origin software.

## Phase separation assays
Protein stock was thawed on ice and diluted up to 200 μM in 20 mM phosphate buffer, pH 7.4. Phase separation of wild-type and G156E FUS-LC was induced by the addition of 250 mM NaCl in the reaction mixture. Spontaneous phase separation of FUS-LC into liquid droplets was indicated by the immediate rise in turbidity upon mixing with salt.

## Turbidity assay
The turbidity of monomeric FUS-LC and phase-separated samples of wild-type and G156E FUS-LC were monitored by measuring absorbance at 350 nm on a Multiskan Go (Thermo Scientific) plate reader. Droplet reactions of 100 μL (200 μM FUS-LC in 20 mM phosphate, 250 mM NaCl, pH 7.4) were set up and used for the turbidity measurements. The mean and standard errors were obtained from at least 3 independent sets of measurements.

## Fluorescence labeling
Single-cysteine FUS-LC variants were labeled with fluorescein-5-maleimide (F-5-M) and AlexaFluor488-C5-maleimide under denaturing buffer conditions (8 M Urea, 20 mM phosphate, pH 7.5) for anisotropy and FRAP measurements. Pure protein was incubated with 0.3 mM tris(2-carboxyethyl)phosphine (TCEP) for 30 min on ice and was mixed with fluorescent dyes in a molar ratio of 1:30 (for F-5-M) and 1:3 (for AlexaFluor488-maleimide). The labeling mixture was incubated in the dark under stirring conditions at room temperature for 3 h. Following the reaction, the excess free dye was removed by buffer exchange using a NAP-10 column. For dual-labeling of FUS-LC single-cysteine variants, the pure protein was incubated under denaturing conditions (8 M Urea, 20 mM phosphate, pH 8) with the amine-reactive NHS ester of AlexaFluor488 (donor dye) in a molar ratio of 1:4 under shaking at 25 °C for 4 h. The unreacted dye was further removed using a NAP-10 column, and the eluted protein was concentrated and used for labeling with the thiol-reactive acceptor dye. The donor-labeled protein was mixed with AlexaFluor594-maleimide in a ratio of 1:4 and incubated at 25 °C with stirring for 5 h, under denaturing conditions (8 M Urea, 20 mM phosphate, pH 7.5). The labeling reaction was then buffer exchanged with a NAP-10 column, and the remaining free dye was removed using a 3 kDa MWCO Amicon filter. All the single and dual-labeled proteins were concentrated using a 3 kDa MWCO Amicon filter. Labeling efficiencies were estimated by measuring absorbance at 280 nm ($\varepsilon_{280nm} = 30,720 \, M^{-1} cm^{-1}$, for FUS-LC cysteine variants), 494 nm ($\varepsilon_{494} = 73,000 \, M^{-1} cm^{-1}$, for AlexaFluor488 and $\varepsilon_{494} = 68,000 \, M^{-1} cm^{-1}$, for F-5-M) and 590 nm ($\varepsilon_{590} = 92,000 \, M^{-1} cm^{-1}$ for AlexaFluor594) to estimate the total protein and labeled protein concentrations. Using our labeling protocol, we obtained a -100% labeling efficiency for the acceptor fluorophore and -80% labeling efficiency for the donor fluorophore.

## Confocal microscopy
All fluorescence microscopy imaging was performed on ZEISS LSM 980 Elyra 7 super-resolution Microscope using a ×63 oil-immersion objective (N.A. 1.4) and a monochrome cooled high-resolution Axio-CamMRm Rev. 3 FireWire(D) camera. Phase separation of 200 μM unlabeled FUS-LC was induced in the presence of 0.1% AlexaFluor488-labeled FUS-LC by the addition of 250 mM NaCl (20 mM phosphate, pH 7.4). Reactions were incubated at room temperature for 5 min and a 5–10 μL sample was placed on a glass coverslip and imaged using a 488 nm laser diode (11.9 mW). For two-color imaging, the droplet reaction was spiked with 0.05% of dual-labeled FUS-LC and imaged using 488 nm and 590 nm excitation sources, respectively. The images were acquired at 1840 × 1840 pixels and 16-bit depth resolution. Airyscan images of the fluorescently labeled droplets were acquired by utilizing the confocal laser scanning microscope via the Airyscan 2 detector equipped with 32 channels (GaAsP). Image processing and analyses were performed on in-built instrument software Zen Blue 3.2 and ImageJ (NIH, Bethesda, USA).

## Fluorescence recovery after photobleaching (FRAP) measurements
FRAP experiments were performed on ZEISS LSM 980 Elyra 7 super-resolution microscope using a ×63 oil-immersion objective (N.A 1.4) and a monochrome cooled high-resolution AxioCamMRm Rev. 3

FireWire(D) camera. A region of 1 μm was bleached inside droplets doped with 0.1% AlexaFluor488-labeled FUS-LC using a 488 nm laser diode. The recovery was recorded using the Zen Blue 3.2 (ZEISS) software. FRAP measurements were performed for at least 8 independent droplets for both wild-type and G156E FUS-LC. Fluorescence recovery curves were normalized, background corrected, and plotted using the Origin software.

## Steady-state fluorescence measurements

Steady-state FRET experiments were performed on a Fluoromax-4 spectrofluorometer (Horiba Jobin Yvon, NJ, USA) using a 1-mm pathlength quartz cuvette. For all the experiments, 50 nM of dual-labeled FUS-LC variants were used. The donor fluorophore (AlexaFluor488) was excited at 494 nm and fluorescence emission was recorded from 515 nm to 700 nm to monitor both donor and acceptor emission spectra.

## Single-droplet FRET imaging by acceptor photobleaching

Phase separation of 200 μM FUS-LC (20 mM phosphate, pH 7.4) was induced by the addition of 250 mM NaCl in salt in the presence of 0.05 % dual-labeled single-cysteine variants of FUS-LC. The droplet reaction was imaged on a ZEISS LSM 980 Elyra 7 super-resolution microscope using a 63x oil-immersion objective (N.A 1.4) and a monochrome-cooled high-resolution AxioCamMRm Rev. 3 FireWire(D) camera. To determine the FRET efficiency, the acceptor present in a dual-labeled droplet was photobleached using a 594 nm laser, and the increase in the donor fluorescence intensity was recorded upon bleaching the acceptor fluorophore. FRET efficiencies were estimated using the Zen Blue 3.2 (ZEISS) software.

## Dynamic light scattering (DLS)

For estimating the hydrodynamic radii of monomeric FUS-LC, a dynamic light scattering instrument (Malvern Zetasizer) was used. All the reaction buffers were filtered using 0.02 μm filters. Monomeric FUS-LC (50 μM in 20 mM phosphate, pH 7.4) in the absence and presence of 250 mM NaCl was used for measurements at room temperature.

## C$_{sat}$ estimation

Droplet reactions (200 μM FUS-LC) were induced by the addition of 250 mM NaCl (in 20 mM sodium phosphate, pH 7.4 buffer) and incubated at 25 °C for 10 min. The reactions were then subjected to ultra-centrifugation at 25 °C, 18,000 × g for 30 min. The supernatant was removed carefully without disturbing the pellet to estimate the dilute phase concentration. The protein saturation concentration ($C_{sat}$) of the dilute phase was estimated by measuring the absorbance at 280 nm ($\varepsilon_{280}$ = 30,720).

## Single-molecule FRET experiments and data analysis

In single-molecule FRET experiments, the ratiometric FRET efficiency (E) for each molecule is recorded from the fluorescence bursts that are separated into donor ($I_D$) and acceptor ($I_A$) signals using the following equation.

$$E = \frac{1}{1 + \left(\frac{I_D}{I_A}\right)\gamma} \tag{1}$$

where γ is a correction factor obtained from different quantum yields of donor and acceptor dyes and the detection efficiencies for donor and acceptor channels.

Single-molecule FRET experiments were performed using a MicroTime 200 time-resolved confocal microscope (PicoQuant) in a pulsed interleaved excitation (PIE) mode. All the single-molecule FRET experiments were performed in 20 mM phosphate buffer, 250 mM NaCl, pH 7.4. Measurements in the monomeric dispersed phase were

performed in the presence of 75–150 pM of dual-labeled FUS-LC, and droplet formation of 200 μM FUS-LC was performed in the presence of 5–10 pM of dual-labeled protein. Data were acquired within 15–20 min after the initiation of the phase separation. Reactions were set in a buffer containing n-propyl gallate as an oxygen scavenger to improve the photostability of the fluorophore in the solution[72]. Pulsed laser sources (485 nm and 594 nm) at a frequency of 20 MHz were used to alternately excite the donor and the acceptor fluorophores within the dual-labeled samples using a ×60 water-immersion objective (N.A. = 1.2). The laser power was fixed at 45–60 μW (40–50 μW for 485 nm and 5–10 μW for 594 nm laser) measured at the back aperture of the objective for the dispersed phase and 5.5–8.5 μW (5–7 μW for 485 nm and 0.5–1.5 μW for 594 nm laser) in order to minimize the saturation, background counts, and photobleaching of the acceptor. The lasers were focused inside the solution (50 μm from the surface) for the dispersed phase and within single droplets (2–4 μm inside) for the condensed phase to obtain fluorescence emission bursts. The emitted photons were collected and focused through a 50 μm pinhole, and a dichroic beam splitter (zt594rdc) was used to separate the donor and acceptor emission. The emission was filtered (BP 535/50 nm for the green channel and LP 594 for the red channel) and detected by the respective single-photon avalanche diode (SPAD) detectors. Data were collected and analyzed using the SymphoTime64 software v2.7. A typical binning time used was 0.5 ms and 1 ms, and using PIE, the bursts containing both donor and acceptor signals were considered for FRET analysis. The donor and acceptor counts were corrected for the background with a minimum threshold of 35 photons used for the further selection of bursts to construct the FRET efficiency histogram. The FRET efficiencies were corrected for the spectral crosstalk between the donor and acceptor fluorophore (α = 0.05), direct excitation of the acceptor by donor laser (β = 0.003), and the correction factor in the detection efficiencies of the donor and acceptor channels (γ = 1.12) which were estimated by performing comparative measurements in both acceptor and donor channels of the instrument and a fluorescence spectrophotometer[73,74]. All the FRET efficiency histograms were constructed for >20,000 events. The FRET efficiency histograms were plotted and fitted using a Gaussian peak function in the Origin software.

## Distance estimation from single-molecule FRET

The inter-dye distance (r) between the N-terminally labeled Alexa-Fluor488 and AlexaFluor594 at a cysteine residue was estimated using the mean FRET efficiency (E) obtained from single-molecule FRET histograms using the following relationships.

$$E = \frac{1}{1 + \left(\frac{r}{R_0}\right)^6} \tag{2}$$

$$r = R_0 \left[\frac{1}{E} - 1\right]^{\frac{1}{6}} \tag{3}$$

For the distance estimation in the monomeric dispersed form, a Förster radius ($R_0$) of 54 Å for AlexaFluor488 and AlexaFluor594 was used[69,75]. For droplets, the $R_0$ was corrected (52.4 Å) by taking the altered index of refraction within condensates into account. We employed the previously reported method for correction by estimating the protein concentration within the protein-rich dense phase[76]. The protein concentration within droplets was estimated by the sedimentation assay by pelleting the dense phase at 25 °C, 18,000 × g for 30 min[59]. Time-resolved fluorescence anisotropy decay measurements for both AlexaFluor488-labeled and AlexaFluor594-labeled FUS-LC, both in monomer and droplets, revealed considerable local fluorophore rotational dynamics validating the assumption of the orientation factor to be $^2/_3$ (Supplementary Fig. 2e, f). However, we would like

to point out here that we do not rule out minor effects of the orientation factor and photon shot noise on inter-dye distances. These effects can be more pronounced in the condensed phase than in the monomeric dispersed phase precluding us from the distance estimation in droplets.

## Fluorescence correlation spectroscopy (FCS)

FCS measurements were performed using the same MicroTime 200 setup for the monomeric phase and the condensed phase of wild-type and G156E FUS-LC. A free dye solution of 1 nM Alexa-Fluor488 was used to estimate the structure parameter of confocal volume (5.52), which was used for further FCS analyses. For the dispersed phase measurements, data were acquired in the presence of 10 nM AlexaFluor488-labeled FUS-LC in the presence of salt (250 mM NaCl). For single-droplet FCS measurements, droplet reactions were set up with 1–3 nM AlexaFluor488 labeled FUS-LC, and these samples were placed on glass coverslips. Measurements were performed by focusing inside the solution for the dispersed phase and within single droplets for the condensed phase of wild-type and G156E FUS-LC. FCS data were collected and analyzed, and correlation curves were fitted with the triplet-state model using SymphoTime64 software v2.7 to obtain diffusion time within the dispersed and condensed phases[53].

## Single-droplet steady-state and time-resolved fluorescence anisotropy measurements

MicroTime 200 time-resolved confocal microscope (PicoQuant) was used for performing steady-state fluorescence anisotropy measurements of dispersed and droplet phase of FUS-LC spiked with 0.1% F-5-M-labeled single-cysteine variants at residue positions 16, 86, 108, and 148. Fluorescein-5-maleimide (F-5-M) dye was used as a thiol-reactive anisotropy probe owing to its short linker length which accurately reports on the rotational dynamics of the polypeptide chain. Freshly phase-separated droplet reactions were spotted on a coverslip with a thickness of 1.5 mm placed directly on a Super Apochromat 60x water immersion objective with 1.2 NA (Olympus). Samples were excited with the 485 nm laser, and the emitted fluorescence was collected and filtered by a bandpass emission filter (BP 535/50) before entering the pinhole (50 μm). The in-focus emitted light exiting the pinhole was split into the two detector channels by a polarizing beam-splitter placed before the detectors and detected by the respective Single-Photon Avalanche Diodes (SPADs). Anisotropy imaging was performed for single-droplet steady-state anisotropy measurements, and a point time trace was obtained for time-resolved anisotropy measurements. The correction factors were calculated by performing fluorescence measurements in a free dye solution and utilized to estimate steady-state anisotropy using the commercially available SymphoTime64 software v2.7. The fluorescence anisotropy ($r_{ss}$) is given by the following relationship.

$$r_{ss} = \frac{I_{||} - I_{\perp}}{[1 - 3L2]I_{||} + [2 - 3L1]I_{\perp}} \quad (4)$$

where $I_{||}$ and $I_{\perp}$ are the background corrected parallel and perpendicular fluorescence intensities, and L1 (0.308) and L2 (0.0368) are the objective correction factors[77].

For time-resolved fluorescence anisotropy decay analysis, the decay profiles obtained from SymphoTime64 software v2.7 were further analyzed by global fitting using the following relationships.

$$I_{||}(t) = 1/3 I(t)[1 + 2r(t)] \quad (5)$$

$$I_{\perp}(t) = 1/3 I(t)[1 - r(t)] \quad (6)$$

where $I_{||}(t)$, $I_{\perp}(t)$, and $I(t)$ denote the time-dependent fluorescence intensities collected at the parallel, perpendicular, and magic angle (54.7°) geometry. The perpendicular component was always corrected using the G-factor that was intendedly obtained from free dye in the buffer. The time-resolved fluorescence anisotropy decay profiles were fitted using a biexponential decay model yielding two rotational correlation times, namely, fast ($\phi_1$) and slow ($\phi_2$) rotational correlation times as follows.

$$r(t) = r_0 \left[ \beta_1 e^{\left(\frac{-t}{\phi_1}\right)} + \beta_2 e^{\left(\frac{-t}{\phi_2}\right)} \right] \quad (7)$$

where $r_0$ denotes the (time-zero) fundamental anisotropy of the fluorophore, and $\beta_1$ and $\beta_2$ the fractional amplitudes associated with $\phi_1$ and $\phi_2$, respectively. The goodness of fit was estimated based on the autocorrelation function, randomness of residuals, and reduced $\chi^2$ values[51].

## Raman spectroscopy

Raman spectra of phase-separated individual droplets of wild-type and G156E FUS-LC were acquired on an inVia laser Raman microscope (Renishaw, UK) at room temperature. Freshly phase-separated samples (3–5 μL) were drop cast onto a glass slide covered with aluminum foil and spectra were obtained within 15–20 min after phase separation initiation. Single droplets were focused using a ×100 long working distance objective lens (Nikon, Japan). The samples were excited with an NIR laser (785 nm) with an exposure time of 10 s at a laser power of 500 mW (100%), and an edge filter of 785 nm was used to block the Rayleigh scattering. The collected Raman scattering was dispersed using a diffraction grating (1200 lines/mm) and further detected by an air-cooled CCD detector. Data were acquired for 5 accumulations, after which collected spectra were background-corrected and smoothened using inbuilt software Wire 3.4. Since a tyrosine vibrational band could potentially interfere with the backbone amide band, amide I bands were separately baseline-corrected and deconvoluted to estimate the secondary structural contents. All the data were plotted and analyzed using the Origin software.

## Reporting summary

Further information on research design is available in the Nature Portfolio Reporting Summary linked to this article.

## Data availability

The data are available within the article, Supplementary Information, or Source data files. Source data are provided with this paper.

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

## Acknowledgements

We thank IISER Mohali, Science and Engineering Research Board (SUPRA SPR/2020/000333 to S.M.), Department of Science and Technology, Govt. of India (FIST grant # SR/FST/LS-II/2017/97 to the Department of Biological Sciences, IISER Mohali), and Ministry of Education, Govt. of India (Centre of Excellence grant to S.M. and the Prime Minister's Research Fellowship to A.W.) for financial support. We thank Professor N. Periasamy (Retd. TIFR Mumbai) for the fluorescence decay analysis program, S. G. Pattanashetty for his valuable contribution during the initial phase of the project, and Dr. M. Bhattacharya (Thapar Institute), Professor A. Deniz (Scripps), Dr. D. Bhowmik (IISER Mohali), and other current and former members of the Mukhopadhyay lab for their valuable input and for critically reading this manuscript.

## Author contributions

A.J. and S.M. conceived the project. A.J., A.W., and S.M. further developed the concept and the experimental design. A.J., A.W., A.A., S.K.R., L.A., and S.S. performed the experiments and analyses. A.J. prepared the figures and wrote the first draft. S.M. supervised the work, wrote/edited the manuscript, obtained funding, and provided the overall direction. All authors discussed the results and commented on the manuscript.

## Competing interests

The authors declare no competing interests.
