## [Peer review file · Nature Communications]

REVIEWER COMMENTS

Reviewer #1 (Remarks to the Author):

In this manuscript by Dr. Mukhopadhyay and co-authors, single-molecule FRET studies coupled with fluorescence correlation spectroscopy; picosecond time-resolved fluorescence anisotropy, and vibrational Raman spectroscopy was used to understand the conformational change of FUS N-terminal Prion-like LCD domain induced by phase separation. The authors showed that there are two sub-populations of FUS conformation that can be resolved by single molecule FRET measurement: a low FRET conformation and a high FRET conformation. LLPS induce a shift from high FRET to low FRET conformation. The authors further showed that the dynamics of the protein, as measured by translational diffusion by FCS and rotational diffusion by time-resolved anisotropy, are both slowed down in liquid droplets. In addition, ALS causing mutation FUS G156E further decrease the dynamics of the liquid droplets. smFRET and time-resolved spectroscopy methods are useful in obtaining protein conformation and dynamic information in the dilute phase and the condensed phase. The manuscript is well written. The main conclusions are supported by the data presented. I have just a few suggestions:

Major concern:

1. The conclusion that different sub-populations of FUS conformation are present in the dilute phase and the condensed phase is very well supported by the smFRET data. However, the interactions underlying the formation of these different conformations would be hard to determine using the current method. Yet, the authors suggest that the compact conformation is mediated by π - π interactions between multiple tyrosine residues and hydrogen bonding between glutamine sidechains, and the extended conformation is due to the proline. To prove these points, extensive mutagenesis studies with these key residues mutated should be done.
2. The experiments with the mutant G156E were done using labeled WT protein as reporter instead of labeled mutant protein. Although using labeled WT protein can still provide some information about the environment in the mutant droplets, it would be more informative if the experiments could be done with labeled mutant protein. If that is not a choice, I would appreciate a discussion on the different information that could be obtained using WT protein in this experiment, comparing to using labeled mutant protein.
3. The LLPS of FUS is driven by interactions between stickers in the C-terminal RGG and the N-terminal Prion-like LCD domain and the LLPS propensity of the N-terminal prion-like LCD is much lower than the full length protein that has the CTD (Wang et al 2018). In this study, the NTD prion-like LCD is used instead of full length protein. Therefore, different interactions might be mediating the LLPS of the NTD, comparing to the full length protein. I would appreciate it if the authors could address this differences in the discussion.

Minor comments:

1. "An archetypal phase-separating protein, Fused in Sarcoma (FUS), is a highly abundant protein belonging to the FUS family of proteins.", in this sentence, it should be FET family of proteins.
2. On page 5, "However, FUS-LC carries a low net charge with an NCPR value", it would be helpful for the reader if the authors could explain what is NCPR.
3. For the FRET efficiency histogram, it would be helpful to report the number of events, instead of probability.
4. In Figure 3E, is the shift of FRET efficiency from 0.64 to 0.76 statistically significant?
5. I am wondering if the fluorescence lifetime changes between the diffuse phase and the condensed phase. If yes, does this change affect the time-resolved anisotropy measurement?
6. In Figure 4A and 5D, the time axis only displays up to 1 second. With the slow diffusion time in the droplets, the baseline and the fitting to the baseline are not able to be seen. I would appreciate it if the figure can display longer correlation time and the fitting of the baseline.
7. In Figure 4e, the anisotropy in the droplets does not decay to 0, as in the monomer condition. This indicates higher rigidity and also different baseline for fitting. However, in the fitting equation, this

shifted baseline is not considered. I am wondering if including a fitting parameter that accounts for baseline shift would improve the accuracy of the fitting. Please see equation 6 in the supplementary document here: <https://www.nature.com/articles/s41557-020-0465-9#Sec15>

8. Page 15, please explain what is catGranule and what does it quantify.

9. The authors show that G156E reduces the LLPS propensity of the Prion-like LCD of FUS. While published results suggest that G156E increase the propensity of FUS aggregation, but does not change the propensity for FUS LLPS (Rhine 2020 and Patel 2015). Is this a difference between full length FUS and the NTD?

Reviewer #2 (Remarks to the Author):

The authors have investigated the structure and properties of FUS-LC molecules in different states (monomeric state and condensed state in a droplet) using various experimental techniques. They also studied the differences in structure and properties between the natural form and a mutant form of FUS-LC with one different amino acid.

The authors examined the fluidity of the molecules through FCS (Fluorescence Correlation Spectroscopy) and FRAP (Fluorescence Recovery After Photobleaching). They also examined the molecular structure or morphology using single-droplet FRET (Förster Resonance Energy Transfer) and single-molecule FRET. Through a single droplet Raman vibrational study, they measured the structural heterogeneity and the degree of hydrogen bonding (the level of exposure to water), demonstrating good compatibility with other results.

In the case of disease-related proteins, they showed even slower translational and rotational motions and a more extended structure (from paperclip-like to partially extended tadpole-like conformers). They also suggested that the proteins form more hydrogen bonds due to higher exposure to water. The authors have tried to elucidate these physical properties. Considering the importance and attention given to the field of liquid-liquid phase separation (LLPS), these experimental results could be of some value. However, the results are somewhat predictable and seem to be previously verified. It is not clear whether these findings are not known formerly. Due to the importance of the topic itself, extensive computational and experimental studies have been conducted on the kinetic arrest and multiphase architecture of FUS liquid droplets during aging. In addition to the references cited in the manuscript (25,31,35,49,63,64), the following two FUS research papers by the Knowles group are other good examples (<https://doi.org/10.1073/pnas.2119800119>, <https://doi.org/10.1101/2022.08.15.503964>).

1. Contrary to the author's claims in Introduction, we have learned little about inter-conversion dynamics. It's clear that there are limitations to obtaining such information with confocal-type single-molecule FRET (smFRET) as the molecules quickly pass through the focal point, as compared to the previous ensemble approaches.

2. Interestingly, replacing just a single amino acid, G156E, seems to make a notable difference. However, the authors did not carefully examine the effect of replacing one amino acid with cysteine for labeling and attaching a bulky fluorescent dye to it. This is an intrinsic limitation of smFRET utilizing bulky donor-acceptor fluorophores. Thus, the main conclusion here is still questionable.

3. There is a brief description of the measurement of labeling efficiency in SI, but it's not clear how efficient it is. Especially, I couldn't find the ratios of unlabeled, singly donor-labeled, singly acceptor-labeled, and dually labeled ones.

4. In the Raman experiments, the wave numbers of the amide I band corresponding to ordered (wild type) and non-regular (G156E) states nearly overlap, and it's unclear how one can confidently assign

them. Such Raman band assignments require more convincing data. In the community of vibrational spectroscopy, including linear and nonlinear IR, Raman, IR-vis SFG, etc, huge efforts have been made to confirm their band assignments with extensive collections of data. However, here the authors simply used two or three bands (Gaussian or Lorentzian) to fit their data and discuss the populations of ordered, non-regular/turn, and non-regular/extended conformers. There would be no vibrational (expert) spectroscopists who agree on such simple band assignments.

5. The authors postulated the pi-pi interaction of tyrosine and hydrogen bonding of glutamine as main interchain contacts of FUS-LC. However, this needs to be corroborated with further evidence other than just PScore data. The authors suggested the intensity changes of Raman spectra (Tyr Fermi doublet) is evidence of interaction between solvent and tyrosine. However, it appears to be just a minor change, considering the error bars. The peaks around 1050 wavenumbers exhibit far more differences, but they were not even mentioned.

6. FUS droplet is a system whose physical properties change through an aging process within a few minutes. It is necessary to clarify the age of the droplets used in the experiment (how much time has passed since phase separation). The aging process of FUS droplets is of great interest in this field. However, the same experiments on droplets of different ages and detailed comparisons are not found in this work.

6. The estimation of structural conformation (paperclip-like and tadpole-like) could have been possible if the authors had considered the distance of D and A with FRET efficiency. The necessary information of the calculated distance between the N-terminus and specific amino acids (C-terminus of 86, 108, and 148) could be found in many previous theoretical studies of the FUS-LC structure as cited.

7. The authors presented the significant meaning of this study. The differences obtained between the natural and disease-related forms are in fluidity, structure, structural distribution, and exposure to water. However, I get the impression that this study's impacts are somewhat exaggerated. Further studies are needed to observe the formation of aggregates, solidification, and aging due to external factors or over time and how the quantities measured in this study change during these processes.

For these reasons, the results and significance of this paper are not sufficiently substantiated. Thus, this reviewer does not recommend its publication in Nat. Comm.

Reviewer #3 (Remarks to the Author):

In this manuscript, Joshi et al. perform fluorescence imaging and vibrational spectroscopy to characterize the conformational dynamics of the low complexity domain of FUS (FUS LC) in the monomeric and liquid droplet states. The main strength of the manuscript is the focus on single molecule studies in both the monomeric and liquid droplet states, which reveals conformational states, fluctuations and interconversion dynamics that are often hard to distinguish in bulk experiments. The main conclusion of the manuscript is that FUS LC exists in two relatively well-defined conformations, a more compact paperclip-like state and a more extended tadpole-like state in the monomeric form. Upon droplet formation, intramolecular interactions get replaced with intermolecular interactions, a process that favors the partially extended tadpole-like conformations. Repeating the experiments with a disease-relevant mutant, G156E, revealed that the mutant is more extended and that it forms a more extensive network of intermolecular interactions in the droplet state. Overall, this study provides valuable information regarding the elusive interactions that drive the phase separation of FUS LC and can serve as a blueprint in performing similar comprehensive single molecule imaging and spectroscopic studies for other phase separating systems. The manuscript is well written and will be of interest to the broader readership of the journal.

However, there are a few things that should be discussed:

The conclusions presented here, namely that the population of partially extended tadpole like states increases in the droplet environment seems to be contrary to the findings of Ref. 35 where a more compact state for FUS LC is favored in the droplet state. Could the authors comment on this potential discrepancy?

FUS LC G156E has been known to undergo a relatively quick liquid to solid transition. How long did the authors follow the single molecule dynamics and Raman signatures in this sample and did they observe any changes in dynamics and structure over time?

Figure 5k – There are some differences in the Raman spectra near 1000 cm^{-1} . Could the authors comment on these differences?

Point-by-point response to reviewers' comments (MS # NCOMMS-23-27076)

Reviewer #1

Reviewer Comments: In this manuscript by Dr. Mukhopadhyay and co-authors, single-molecule FRET studies coupled with fluorescence correlation spectroscopy; picosecond time-resolved fluorescence anisotropy, and vibrational Raman spectroscopy was used to understand the conformational change of FUS N-terminal Prion-like LCD domain induced by phase separation. The authors showed that there are two sub-populations of FUS conformation that can be resolved by single molecule FRET measurement: a low FRET conformation and a high FRET conformation. LLPS induce a shift from high FRET to low FRET conformation. The authors further showed that the dynamics of the protein, as measured by translational diffusion by FCS and rotational diffusion by time-resolved anisotropy, are both slowed down in liquid droplets. In addition, ALS causing mutation FUS G156E further decrease the dynamics of the liquid droplets. smFRET and time-resolved spectroscopy methods are useful in obtaining protein conformation and dynamic information in the dilute phase and the condensed phase. The manuscript is well written. The main conclusions are supported by the data presented. I have just a few suggestions.

Authors' Response: We thank the reviewer for her/his kind words, appreciation, and insightful comments on our work. Our responses are as follows. The changes in the revised manuscript are marked in blue.

Reviewer Comments: The conclusion that different sub-populations of FUS conformation are present in the dilute phase and the condensed phase is very well supported by the smFRET data. However, the interactions underlying the formation of these different conformations would be hard to determine using the current method. Yet, the authors suggest that the compact conformation is mediated by π - π interactions between multiple tyrosine residues and hydrogen bonding between glutamine sidechains, and the extended conformation is due to the proline. To prove these points, extensive mutagenesis studies with these key residues mutated should be done.

Authors' Response: We agree with the reviewer and thank the reviewer for this suggestion. Our single-molecule FRET studies directly capture the changes in the conformational characteristics of FUS-LC during its phase separation. Previous studies on phase separation of LC (Ref. 25. Murthy et al. *Nat. Struct. Mol. Biol.* 2019, **26**, 637-648) have demonstrated the crucial role of hydrogen bonding and π - π stacking interactions between glutamine and tyrosine residues, respectively, in driving the phase separation. These studies demonstrated that large-scale mutations of these residues considerably altered the phase behavior of LC and can even lead to a complete loss of phase separation propensity. We have now discussed this on pages 10 and 14 in our revised manuscript. In a separate work, we are pursuing a large-scale mutational study to discern the role of sequence composition and posttranslational modifications on the conformation of the monomeric FUS-LC and will report these results in due course.

Reviewer Comments: The experiments with the mutant G156E were done using labeled WT protein as reporter instead of labeled mutant protein. Although using labeled WT protein can still provide some information about the environment in the mutant droplets, it would be more informative if the experiments could be done with labeled mutant protein. If that is not a choice, I would appreciate a discussion on the different information that could be obtained using WT protein in this experiment, comparing to using labeled mutant protein.

Authors' Response: We thank the reviewer for asking this question. As we stated on page 11, we aimed to study the altered conformational behavior of the labeled LC wildtype within condensates derived from the disease-associated mutant (G156E). Therefore, we used a picomolar concentration of FRET-pair-labeled LC wildtype as a reporter of the environment inside the condensates. This FRET-pair-labeled reporter protein readjusts its conformational distribution based on the internal environment, extent of interactions, and conformational distributions of the densely packed surrounding polypeptide chains within the condensates of G156E FUS-LC. The sensitivity of single-molecule FRET allowed us to record the changes in conformational distribution within these condensates. We have now discussed this in detail on Page 11 in the Results section of our revised manuscript.

Reviewer Comments: The LLPS of FUS is driven by interactions between stickers in the C-terminal RGG and the N-terminal Prion-like LCD domain and the LLPS propensity of the N-terminal prion-like LCD is much lower than the full length protein that has the CTD (Wang et al 2018). In this study, the NTD prion-like LCD is used instead of full length protein. Therefore, different interactions might be mediating the LLPS of the NTD, comparing to the full length protein. I would appreciate it if the authors could address this differences in the discussion.

Authors' Response: This is a very nice suggestion. We have now discussed these differences in detail on Page 14 in the Discussion section of our revised manuscript.

Reviewer Comments: “An archetypal phase-separating protein, Fused in Sarcoma (FUS), is a highly abundant protein belonging to the FUS family of proteins.”, in this sentence, it should be FET family of proteins.

Authors' Response: We have now changed it to “FET” in the revised manuscript.

Reviewer Comments: On page 5, “However, FUS-LC carries a low net charge with an NCPR value”, it would be helpful for the reader if the authors could explain what is NCPR.

Authors' Response: We agree with the reviewer and thank her/him for raising this point. We have now briefly explained NCPR on Page 4 in the Results section of our revised manuscript.

Reviewer Comments: For the FRET efficiency histogram, it would be helpful to report the number of events, instead of probability.

Authors' Response: We thank the reviewer for this suggestion. We have reported the probability in the FRET efficiency histograms to maintain uniformity and simplify the plots. In our revised manuscript, we have also included the number of events in the figure legends for all of the histograms.

Reviewer Comments: In Figure 3E, is the shift of FRET efficiency from 0.64 to 0.76 statistically significant?

Authors' Response: We thank the reviewer for raising this point; this shift is statistically significant at $P < 0.005$ (***).

Reviewer Comments: I am wondering if the fluorescence lifetime changes between the diffuse phase and the condensed phase. If yes, does this change affect the time-resolved anisotropy measurement?

Authors' Response: This is an interesting question. The steady-state anisotropy is dependent on the fluorescence lifetime (Perrin's relationship), whereas, the rotational correlation times recovered from time-resolved anisotropy decays are independent of fluorescence lifetime. In our case, we see no

significant changes in the fluorescence lifetime in the dilute and the dense phase. This is now stated in Figure 4d legend.

Reviewer Comments: In Figure 4A and 5D, the time axis only displays up to 1 second. With the slow diffusion time in the droplets, the baseline and the fitting to the baseline are not able to be seen. I would appreciate it if the figure can display longer correlation time and the fitting of the baseline.

Authors' Response: We thank the reviewer for pointing this out. We have now changed the time axes in the plots to display longer correlation times and the fitting of the baseline.

Reviewer Comments: In Figure 4e, the anisotropy in the droplets does not decay to 0, as in the monomer condition. This indicates higher rigidity and also different baseline for fitting. However, in the fitting equation, this shifted baseline is not considered. I am wondering if including a fitting parameter that accounts for baseline shift would improve the accuracy of the fitting. Please see equation 6 in the supplementary document here: <https://www.nature.com/articles/s41557-020-0465-9#Sec15>

Authors' Response: This is an interesting point raised by the reviewer. The model that is described in the aforesaid paper is based on the hindered rotor model utilizing the residual anisotropy (r_∞) that is typically used to explain the restricted rotational dynamics (wobbling-in-cone motion) of a fluorophore in the rigid gel phase of the membrane. In our case, we did not use the hindered rotor model since the droplet interior is mobile which is also evident by the slope of the anisotropy decay in the longer time regime. This slope allows us to estimate the slow rotational correlation time from the global analysis with reasonable accuracy.

Reviewer Comments: Page 15, please explain what is catGranule and what does it quantify.

Authors' Response: We thank the reviewer for asking this question. We have now included a brief description of catGranule on Pages 10 and 11 in the Results section of our revised manuscript.

Reviewer Comments: The authors show that G156E reduces the LLPS propensity of the Prion-like LCD of FUS. While published results suggest that G156E increase the propensity of FUS aggregation, but does not change the propensity for FUS LLPS (Rhine 2020 and Patel 2015). Is this a difference between full length FUS and the NTD?

Authors' Response: This is an important point. Although the phase separation propensity of the full-length FUS appears to remain similar upon the G156E mutation, the mutant exhibits an accelerated aggregation and possibly dynamically arrested physical property as reported by Rhine et al. *Mol. Cell* **19**, 666-681(2020). In the case of the FUS-LC, the introduction of G156E mutation slightly decreases the phase separation propensity, as evident by our C_{sat} experiments and catGranule prediction tool, possibly due to an additional negative charge facilitating an enhanced solubility. Thus, the G156E mutation exhibits different effects on the phase separation of the full-length and low-complexity domain of FUS. We have now included this on Page 11 in the Results section of our revised manuscript.

We are extremely grateful to this reviewer for her/his valuable comments and suggestions that helped us improve our manuscript.

Reviewer #2

Reviewer Comments: The authors have investigated the structure and properties of FUS-LC molecules in different states (monomeric state and condensed state in a droplet) using various experimental techniques. They also studied the differences in structure and properties between the natural form and a mutant form of FUS-LC with one different amino acid.

The authors examined the fluidity of the molecules through FCS (Fluorescence Correlation Spectroscopy) and FRAP (Fluorescence Recovery After Photobleaching). They also examined the molecular structure or morphology using single-droplet FRET (Förster Resonance Energy Transfer) and single-molecule FRET. Through a single droplet Raman vibrational study, they measured the structural heterogeneity and the degree of hydrogen bonding (the level of exposure to water), demonstrating good compatibility with other results.

In the case of disease-related proteins, they showed even slower translational and rotational motions and a more extended structure (from paperclip-like to partially extended tadpole-like conformers). They also suggested that the proteins form more hydrogen bonds due to higher exposure to water. The authors have tried to elucidate these physical properties. Considering the importance and attention given to the field of liquid-liquid phase separation (LLPS), these experimental results could be of some value. However, the results are somewhat predictable and seem to be previously verified. It is not clear whether these findings are not known formerly. Due to the importance of the topic itself, extensive computational and experimental studies have been conducted on the kinetic arrest and multiphase architecture of FUS liquid droplets during aging. In addition to the references cited in the manuscript (25,31,35,49,63,64), the following two FUS research papers by the Knowles group are other good examples (<https://doi.org/10.1073/pnas.2119800119>, <https://doi.org/10.1101/2022.08.15.503964>).

Authors' Response: We thank the reviewer for insightful comments and suggestions. Our responses are as follows. The changes in the revised manuscript are marked in blue. We would like to emphasize that our work utilizes the unique capability of single-molecule FRET to directly observe the conformational distributions in the monomeric dispersed phase and the condensed phase. We adapted and utilized the single-molecule technology to be able to record single-molecule fluorescence bursts within individual condensates. Using our novel experimental design and strategy, we were able to detect the key structural transitions associated with phase separation. Such direct observations are generally skewed in conventional ensemble experiments due to the inherent averaging issue. On the contrary, our single-molecule FRET experiments clearly discern the conformational distributions and subpopulations on going from a dispersed phase to a condensed phase of a well-studied prion-like low-complexity domain that serves as the benchmark in the phase separation field. We believe that our approach will now open new and exciting avenues to characterize biological phase transitions of a wide range of proteins at the single-molecule resolution by monitoring dynamic structural distributions in a molecule-by-molecule fashion. Such studies will be of immense importance in the field and will push the boundaries of condensate biology.

Reviewer Comments: Contrary to the author's claims in Introduction, we have learned little about inter-conversion dynamics. It's clear that there are limitations to obtaining such information with confocal-type single-molecule FRET (smFRET) as the molecules quickly pass through the focal point, as compared to the previous ensemble approaches.

Authors' Response: We thank the reviewer for asking this question. Our single-molecule FRET studies illuminate the conformational distribution and interconversion dynamics of a low-complexity domain

that undergoes phase separation. In-and-out diffusion of FRET-labeled molecules through the confocal volume allowed us to record the conformational states of individual freely diffusing molecules that are conformationally stable on the diffusion timescale. Using a varied binning time, we were able to show that the structural subpopulations interconvert slower than a millisecond. We have now included in the Discussion on page 14, “Such slower conformational exchanges between extended and compact states can further be studied using surface-immobilized single-molecule FRET.”

Reviewer Comments: Interestingly, replacing just a single amino acid, G156E, seems to make a notable difference. However, the authors did not carefully examine the effect of replacing one amino acid with cysteine for labeling and attaching a bulky fluorescent dye to it. This is an intrinsic limitation of smFRET utilizing bulky donor-acceptor fluorophores. Thus, the main conclusion here is still questionable.

Authors' Response: We thank the reviewer for raising this concern; however, we respectfully disagree with this statement. We would like to note that the glycine-to-glutamate mutation is a significantly perturbing mutation both in terms of conformational flexibility and the net charge. It is a well-studied disease-associated mutant that is known to drastically alter the aggregation behavior of FUS. On the contrary, the introduction of a cysteine residue by replacing a serine or an alanine residue is not known to have such significant consequences on the polypeptide chain. Previous studies (Murthy, A. C., et al. *Nat. Struct. Mol. Biol.* 2019, **26**, 637–648) have also used similar mutants to characterize the structural ensemble of FUS-LC using NMR. These studies didn't find any discernable changes in the structural propensity. Additionally, rich literature in single-molecule FRET indicates that labeling using polar, water-soluble, Alexa dyes containing a flexible linker does not significantly affect the conformational characteristics of highly flexible IDPs.

Reviewer Comments: There is a brief description of the measurement of labeling efficiency in SI, but it's not clear how efficient it is. Especially, I couldn't find the ratios of unlabeled, singly donor-labeled, singly acceptor-labeled, and dually labeled ones.

Authors' Response: We thank the reviewer for pointing this out. We would like to mention that in our studies, we used the more advanced Pulsed-Interleaved Excitation FRET (PIE-FRET) mode for single-molecule data acquisition and analysis which is typically used to eliminate the effect of singly labeled molecules and selects only the dual-labeled species for constructing the single-molecule FRET histograms. In any case, we have now included the details about labeling efficiencies on Page S4 in the Supplementary Methods section.

Reviewer Comments: In the Raman experiments, the wave numbers of the amide I band corresponding to ordered (wild type) and non-regular (G156E) states nearly overlap, and it's unclear how one can confidently assign them. Such Raman band assignments require more convincing data. In the community of vibrational spectroscopy, including linear and nonlinear IR, Raman, IR-vis SFG, etc, huge efforts have been made to confirm their band assignments with extensive collections of data. However, here the authors simply used two or three bands (Gaussian or Lorentzian) to fit their data and discuss the populations of ordered, non-regular/turn, and non-regular/extended conformers. There would be no vibrational (expert) spectroscopists who agree on such simple band assignments.

Authors' Response: The amide I band for the wild-type FUS-LC droplets upon deconvolution yields three bands at 1652, 1671, and 1692 cm^{-1} . While for the disease mutant, the amide I band gives rise to two Gaussian bands at 1666 and 1692 cm^{-1} . The extensive protein Raman work by several experts in the field has allowed the assignment of the band at 1671 cm^{-1} to β -sheet conformation while the band at 1692 cm^{-1} has been assigned to extended/unordered structures. Additionally, several previous reports

have assigned the band at 1666 cm^{-1} to non-regular/disordered structures. Based on these previous studies, we have assigned the bands as described in the manuscript and have now added these references (Supplementary References 12-17) in our revised Supplementary Information.

Reviewer Comments: The authors postulated the pi-pi interaction of tyrosine and hydrogen bonding of glutamine as main interchain contacts of FUS-LC. However, this needs to be corroborated with further evidence other than just PScore data. The authors suggested the intensity changes of Raman spectra (Tyr Fermi doublet) is evidence of interaction between solvent and tyrosine. However, it appears to be just a minor change, considering the error bars. The peaks around 1050 wavenumbers exhibit far more differences, but they were not even mentioned.

Authors' Response: The role of π - π interactions and hydrogen bonding by tyrosine and glutamine residues have been observed in previous computational and ensemble experimental studies of FUS-LC phase separation (Murthy, A. C., et al. *Nat. Struct. Mol. Biol.* 2019, **26**, 637–648) corroborating the PScore data. This further validates our hypothesis that this dense network of transient intermolecular interactions results in a viscoelastic network fluid within the condensates as demonstrated by our picosecond time-resolved fluorescence anisotropy decay and FCS results. We have now explained it clearly on Page 10 in the Results section of our revised manuscript. The changes in the ratio of tyrosine Fermi doublet intensities are small but statistically significant at $P < 0.05$. We would like to note here that this is an intensity ratio that exhibits a more reliable change as opposed to just the intensity changes. To answer the question about the peak around 1050 cm^{-1} wavenumbers, this peak ($\sim 1046\text{ cm}^{-1}$) originates from the proline residues and the intensity variation within the wild-type and G156E FUS-LC droplets can be accounted for the possible difference in their dense phase concentrations and the resulting variation in the amide I normalized spectra. We have cited some of these relevant Raman spectroscopy papers (Supplementary References 12-17) in our revised Supplementary Information.

Reviewer Comments: FUS droplet is a system whose physical properties change through an aging process within a few minutes. It is necessary to clarify the age of the droplets used in the experiment (how much time has passed since phase separation). The aging process of FUS droplets is of great interest in this field. However, the same experiments on droplets of different ages and detailed comparisons are not found in this work.

Authors' Response: We thank the reviewer for raising this important question. Solid-like aggregates formed upon aging are harder to study by single-molecule FRET since the internal protein diffusion is considerably frozen. Full-length FUS droplets tend to undergo fast aging into solid-like structures as also shown by a recent study (Chen, Y. S. A., et al. *Proc. Natl. Acad. Sci. USA* 120 (33), e2301366120 (2023)). However, in the case of the FUS-LC condensates, the maturation and coarsening occur on a much longer timescale ($\sim 48\text{ h}$ for wild-type and $\sim 24\text{ h}$ for G156E FUS-LC) (Berkeley R. F., et al. *Biophys J.* 2021, 120(7), 1276-1287). In our work, data acquisition for droplets was performed within 15-20 minutes after initiation of phase separation. We have now mentioned the time of data acquisition on Pages S7 and S9 in the Supplementary Information.

Reviewer Comments: The estimation of structural conformation (paperclip-like and tadpole-like) could have been possible if the authors had considered the distance of D and A with FRET efficiency. The necessary information of the calculated distance between the N-terminus and specific amino acids (C-terminus of 86, 108, and 148) could be found in many previous theoretical studies of the FUS-LC structure as cited.

Authors' Response: We thank the reviewer for this suggestion. The FRET efficiency histograms provide evidence of conformational heterogeneity and subpopulations in a direct model-free fashion. An important takeaway from our work is the direct observation of conformational unwinding during phase separation allowing the formation of a multivalent network. The estimation of exact distances is not critically important in our study especially, since such estimations are complicated by certain approximations involving the orientation factor and the photon shot noise as described in the Discussion section on page 15 of our revised manuscript.

Reviewer Comments: The authors presented the significant meaning of this study. The differences obtained between the natural and disease-related forms are in fluidity, structure, structural distribution, and exposure to water. However, I get the impression that this study's impacts are somewhat exaggerated. Further studies are needed to observe the formation of aggregates, solidification, and aging due to external factors or over time and how the quantities measured in this study change during these processes.

Authors' Response: We respectfully disagree with the reviewer. Using a novel experimental design and a unique strategy involving single-droplet single-molecule FRET in conjunction with FCS, picosecond time-resolved fluorescence anisotropy, and Raman spectroscopy, we have been able to unmask the molecular details of phase separation of an archetypal low-complexity prion-like domain. Our study provides a direct observation of conformational unwinding events that switch the intramolecular interactions into intermolecular contacts giving rise to a network of physical crosslinks ensuing biomolecular condensation. To the best of our knowledge, this is the first direct demonstration of such molecular events in protein phase separation. As we stated above, this work opens new and exciting avenues to characterize biological phase transitions utilizing the unique capabilities of single-molecule technologies coupled with existing and emerging molecular and cellular tools. We believe this work is a stepping stone for the development of more complex and highly sophisticated multi-color, multi-parameter, super-resolved, single-droplet single-molecule FRET that will offer unprecedented spatiotemporal resolution in delineating precise molecular drivers of biological phase transitions involved in physiology and disease.

We are extremely grateful to this reviewer for her/his valuable comments and suggestions that helped us improve our manuscript.

Reviewer #3

Reviewer Comments: In this manuscript, Joshi et al. perform fluorescence imaging and vibrational spectroscopy to characterize the conformational dynamics of the low complexity domain of FUS (FUS LC) in the monomeric and liquid droplet states. The main strength of the manuscript is the focus on single molecule studies in both the monomeric and liquid droplet states, which reveals conformational states, fluctuations and interconversion dynamics that are often hard to distinguish in bulk experiments. The main conclusion of the manuscript is that FUS LC exists in two relatively well-defined conformations, a more compact paperclip-like state and a more extended tadpole-like state in the monomeric form. Upon droplet formation, intramolecular interactions get replaced with intermolecular interactions, a process that favors the partially extended tadpole-like conformations. Repeating the experiments with a disease-relevant mutant, G156E, revealed that the mutant is more extended and that it forms a more extensive network of intermolecular interactions in the droplet state. Overall, this study

provides valuable information regarding the elusive interactions that drive the phase separation of FUS LC and can serve as a blueprint in performing similar comprehensive single molecule imaging and spectroscopic studies for other phase separating systems. The manuscript is well written and will be of interest to the broader readership of the journal.

However, there are a few things that should be discussed:

Authors' Response: We thank the reviewer for her/his kind words, appreciation, and valuable suggestions on our work. Our responses are as follows. The changes in the revised manuscript are marked in blue.

Reviewer Comments: The conclusions presented here, namely that the population of partially extended tadpole like states increases in the droplet environment seems to be contrary to the findings of Ref. 35 where a more compact state for FUS LC is favored in the droplet state. Could the authors comment on this potential discrepancy?

Authors' Response: We thank the reviewer for raising this interesting point. In the work by Kato *et al.* (*Proc Natl Acad Sci U S A.* 2021), using a phase-separated gel-like state of the C-terminal part of the FUS-LC domain (residues 111-214), they have identified a secondary cross- β core (residues 155-190) necessary for hydrogel binding of the monomeric FUS-LC. Conversely, our study focuses on the liquid-like droplets of the FUS-LC domain (residues 1-163) prior to their liquid-to-gel transition. This could be a potential reason for the discrepancy regarding the predominance of partially extended states in the condensed phase in contrast to the ordered structures mentioned in Ref. 36. We have now included these observations in relevance to our studies on Page 14 in the Discussion section of our revised manuscript.

Reviewer Comments: FUS LC G156E has been known to undergo a relatively quick liquid to solid transition. How long did the authors follow the single molecule dynamics and Raman signatures in this sample and did they observe any changes in dynamics and structure over time?

Authors' Response: We thank the reviewer for raising this question. We obtained the single-molecule and Raman data within 15-20 minutes after initiation of phase separation where we did not observe any change in dynamics and structure of the condensates during this time interval. This is in accordance with the findings of a previous study (Berkeley R. F., *et al. Biophys J.* 2021, 120, 1276-1287) which demonstrates the maturation and coarsening of these droplets on a much longer timescale (~ 48 h for wild-type FUS-LC and ~ 24 h for G156E FUS-LC). We have now mentioned the measurement time interval on Pages S7 and S9 in the Supplementary Information (Supplementary Methods) section of our revised manuscript.

Reviewer Comments: Figure 5k – There are some differences in the Raman spectra near 1000 cm^{-1} . Could the authors comment on these differences?

Authors' Response: We thank the reviewer for pointing this out. The peak ~ 990 cm^{-1} originates from the dibasic phosphate group of the buffer. Whereas, the peak ~ 1046 cm^{-1} is attributed to the proline residues. The differences in these peak intensities might indicate a change in the concentration of buffer components (~ 990 cm^{-1}) and the difference in the dense phase concentration within wild-type and G156E FUS-LC droplets. We have now added the band assignment in our revised Supplementary Information (Supplementary Table 5 and Supplementary References 12-17).

We are extremely grateful to this reviewer for her/his valuable comments and suggestions that helped us improve our manuscript.

REVIEWER COMMENTS

Reviewer #1 (Remarks to the Author):

In the revised manuscript, the authors have successfully addressed all my concerns. I think the manuscript is suitable for publication in its current form.

Reviewer #2 (Remarks to the Author):

Although the authors put efforts into addressing the issues raised in the first round review, I am not entirely convinced that this work merits publication in Nat. Comm.

Novelty problem: The main issue

It still remains difficult for me to agree that the novelty of the single droplet SM-FRET method warrants its publication in Nat. Comm.; specifically, the key unresolved points include (a) what distinguishes the authors' approach from conventional SM-FRET, and is it notably novel? and (b) What scientific significance does the author derive from their experiments, which has not been previously uncovered?

In response to our comments, which recommended assessing structural conformations (such as paperclip-like and tadpole-like) by taking into account the FRET efficiency between donor (D) and acceptor (A) and other theoretical investigations if necessary, the authors insisted that estimating distances is not pivotal in their study. They argued that this is especially the case because it would entail complex approximations.

As far as I understand, the beauty of the FRET method lies in its ability to gauge the size of the molecule through the measurement of the distance between D and A, which is why it's often referred to as a 'molecular ruler'. While precise length changes may be challenging to determine, the authors could put an effort into investigating to assess whether the approximate length changes that can be estimated through FRET position changes align with the structural changes claimed in the paper (folding and unfolding).

If the length measurement and (or) its correlation to the structural changes in the molecules are not the primary purpose of the method in this context, it raises the question of why the authors chose to use FRET as the primary tool for the study, despite the existence of various other methodologies, such as Raman spectroscopy, and what advantages it offers over these alternative approaches. Furthermore, it appears that similar conclusions could be more routinely derived using NMR (<https://doi.org/10.1016/j.jmr.2022.107318>).

More importantly, regarding novelty in scientific findings, it seems that similar FRET studies to those presented in this manuscript have already been published in journals such as Cell Reports Methods (2022, <https://doi.org/10.1016/j.crmeth.2022.100184>) and JACS (2021, <https://doi.org/10.1021/jacs.1c03078>), to name a few.

In response to my comment on the lack of novelty in their work, the authors said, "We would like to emphasize that our work utilizes the unique capability of single-molecule FRET to directly observe the conformational distributions in the monomeric dispersed phase and the condensed phase. We adapted and utilized the single-molecule technology to be able to record single-molecule fluorescence bursts within individual condensates".

However, in the paper by Wen et al., (Conformational Expansion of Tau in Condensates Promotes Irreversible Aggregation. J. Am. Chem. Soc. 25, 13056-13064 (2021)), a nearly identical experimental

study was performed. In this JACS paper, there are statements "We then performed confocal smFRET experiments to detect the interdomain conformations of Tau variants under non-LLPS and LLPS conditions in the presence of 15% PEG. Fluorescence donor and acceptor bursts, originating from individual labeled protein molecules, were detected and analyzed to generate FRET efficiency histograms. Note that the peak around zero FRET efficiency ("zero peak") is generally ascribed to molecules that are in an acceptor-inactive state or only carry the donor fluorophore." Therefore, I believe that the authors of the submitted manuscript have not provided any new insights, even though the authors kept emphasizing that their approach and work are "unique," "novel," "first," and so on in their manuscript and response letter.

Hence, I suggest that this study is not the best fit for Nature Communications, which is a multidisciplinary journal with a broad readership. Instead, we advise the authors to contemplate submitting this manuscript to a more specialized journal, where it might be better received by the audience with an appreciation for its content.

Interconversion dynamics:

As mentioned by the authors in their response letter, the authors didn't observe interconversion dynamics directly. Instead, they were able to observe the changes in the populations of the two states. They further mentioned, "Such slower conformational exchanges between extended and compact states can further be studied using surface-immobilized single-molecule FRET." The authors emphasized that the conformational exchange is too slow to be directly observed. However, this is not a relevant answer to this reviewer's question. I pointed out that the diffusive motion of objectives could be fast so that they can diffuse out of the focal point rapidly, which makes it difficult to investigate the interconversion dynamics in real time directly.

Disease model:

The authors used the following expression in their response to my comment: "To the best of our knowledge, this is the first direct demonstration of such molecular events in protein phase separation. As we stated above, this work opens new and exciting avenues to characterize biological phase transitions utilizing the unique capabilities of single-molecule technologies coupled with existing and emerging molecular and cellular tools".

However, it is still necessary to compare and analyze disease models in a piecemeal manner, as they are so important. If the need for additional experiments is suggested, it should be answered by including it in the future research direction. However, the above expression is not valid unless the authors' group was the first to do it, as mentioned in the above (see Novelty problem: Main issue).

Reviewer #3 (Remarks to the Author):

The revisions have addressed my original concerns. Upon rereading the revised manuscript, I found a few confusing statements and figures that the authors may wish to address:

- p.5, the following statement is confusing: "The FRET donor (AlexaFluor488) was installed using the N-terminal NHS chemistry and the acceptor (as stated above), whereas, thiol-active AlexaFluor594-maleimide, was covalently linked ..." It is not clear whether the donor or acceptor was installed by the NHS chemistry.

- p.16 - "FUS-family proteins" should be "FET-family proteins"

- Fig. 5I,m - Which parts of the Raman spectra in k are shown here? Looking at k, it seems that the left side of the spectrum does not go to zero at 1640 cm⁻¹ but instead overlaps with another peak. In

l,m, the intensity at 1640 cm⁻¹ goes to "zero", similar to the right side of the spectrum.

- Fig. 5I - The third component of the fit (the small peak between 1640 and 1660 cm⁻¹) is hard to see. Is it supposed to represent the nonregular/turns conformations?

Point-by-point response to reviewers' comments (NCOMMS-23-27076A)

Reviewer #1

Reviewer Comments: In the revised manuscript, the authors have successfully addressed all my concerns. I think the manuscript is suitable for publication in its current form.

Authors' Response: We thank this reviewer for her/his kind words, appreciation, and valuable suggestions in the previous revision.

Reviewer #2

Reviewer Comments: It still remains difficult for me to agree that the novelty of the single droplet SM-FRET method warrants its publication in Nat. Comm.; specifically, the key unresolved points include (a) what distinguishes the authors' approach from conventional SM-FRET, and is it notably novel? and (b) What scientific significance does the author derive from their experiments, which has not been previously uncovered?

In response to our comments, which recommended assessing structural conformations (such as paperclip-like and tadpole-like) by taking into account the FRET efficiency between donor (D) and acceptor (A) and other theoretical investigations, if necessary, the authors insisted that estimating distances is not pivotal in their study. They argued that this is especially the case because it would entail complex approximations. If the length measurement and (or) its correlation to the structural changes in the molecules are not the primary purpose of the method in this context, it raises the question of why the authors chose to use FRET as the primary tool for the study, despite the existence of various other methodologies, such as Raman spectroscopy, and what advantages it offers over these alternative approaches. Furthermore, it appears that similar conclusions could be more routinely derived using NMR (<https://doi.org/10.1016/j.jmr.2022.107318>).

More importantly, regarding novelty in scientific findings, it seems that similar FRET studies to those presented in this manuscript have already been published in journals such as Cell Reports Methods (2022, <https://doi.org/10.1016/j.crmeth.2022.100184>) and JACS (2021, <https://doi.org/10.1021/jacs.1c03078>), to name a few.

In response to my comment on the lack of novelty in their work, the authors said, "We would like to emphasize that our work utilizes the unique capability of single-molecule FRET to directly observe the conformational distributions in the monomeric dispersed phase and the condensed phase. We adapted and utilized the single-molecule technology to be able to record single-molecule fluorescence bursts within individual condensates".

However, in the paper by Wen et al., (Conformational Expansion of Tau in Condensates Promotes Irreversible Aggregation. J. Am. Chem. Soc. 25, 13056-13064 (2021)), a nearly identical experimental study was performed. In this JACS paper, there are statements "We then performed confocal smFRET experiments to detect the interdomain conformations of Tau variants under non-LLPS and LLPS conditions in the presence of 15% PEG. Fluorescence donor and acceptor bursts, originating from individual labeled protein molecules, were detected and analyzed to generate FRET efficiency histograms. Note that the peak around zero FRET efficiency ("zero peak") is generally ascribed to molecules that are in an acceptor-inactive state or only carry the donor fluorophore." Therefore, I believe that the authors of the submitted manuscript have not provided any new insights, even though the authors

kept emphasizing that their approach and work are “unique,” “novel,” “first,” and so on in their manuscript and response letter.

Interconversion dynamics: As mentioned by the authors in their response letter, the authors didn't observe interconversion dynamics directly. Instead, they were able to observe the changes in the populations of the two states. They further mentioned, “Such slower conformational exchanges between extended and compact states can further be studied using surface-immobilized single-molecule FRET.” The authors emphasized that the conformational exchange is too slow to be directly observed. However, this is not a relevant answer to this reviewer's question. I pointed out that the diffusive motion of objectives could be fast so that they can diffuse out of the focal point rapidly, which makes it difficult to investigate the interconversion dynamics in real time directly.

Disease model: The authors used the following expression in their response to my comment: “To the best of our knowledge, this is the first direct demonstration of such molecular events in protein phase separation. As we stated above, this work opens new and exciting avenues to characterize biological phase transitions utilizing the unique capabilities of single-molecule technologies coupled with existing and emerging molecular and cellular tools”. However, it is still necessary to compare and analyze disease models in a piecemeal manner, as they are so important. If the need for additional experiments is suggested, it should be answered by including it in the future research direction. However, the above expression is not valid unless the authors' group was the first to do it, as mentioned in the above (see Novelty problem: Main issue).

Authors' Response: We respectfully disagree with this reviewer. In our previous response file, we already highlighted the novelty, importance, and significance of our work. This reviewer is asking similar questions and appears to fail to recognize that conventional ensemble measurements do not allow one to obtain the structural distribution in (low) monomeric protein concentrations. The Cell Reports Methods paper the reviewer stated is on ensemble FRET imaging of single foci using fluorescent proteins and is not relevant to our work. The other paper (already cited in our manuscript) did not describe single-molecule FRET in the pulsed-interleaved excitation (PIE) format within individual condensates. Performing single-droplet single-molecule FRET has the unique advantage of recording the conformational distribution only in the protein-rich condensed phase (without contributions from the light phase), and our observed FRET distributions do not include the conformers from the light phase. Additionally, the PIE format allows us to clearly observe the low FRET (expanded) populations since the donor-only events (that give rise to a zero-peak) are not considered for registering as FRET events. In order to further address the other points, we have now included the distance estimation from the mean FRET efficiency that is included on Page 7 in the revised manuscript and Supplementary Information. We have also included a few more sentences on future directions on mutational and post-translational modifications studies on Page 16 in the Discussion sections of our revised manuscript.

Reviewer #3

Reviewer Comments: The revisions have addressed my original concerns. Upon rereading the revised manuscript, I found a few confusing statements and figures that the authors may wish to address: p.5, the following statement is confusing: "The FRET donor (AlexaFluor488) was installed using the N-terminal NHS chemistry and the acceptor (as stated above), whereas, thiol-active AlexaFluor594-maleimide, was covalently linked ..." It is not clear whether the donor or acceptor was installed by the NHS chemistry.

Authors' Response: We are most grateful to this reviewer for helping us improve the manuscript. The FRET donor (AlexaFluor488) was installed using N-terminal NHS chemistry, whereas, the acceptor (AlexaFluor594-maleimide) was covalently linked using thiol-maleimide chemistry as indicated in Figure 1c. We have now changed the sentence on Page 5 in the Results section of our revised manuscript.

Reviewer Comments: p.16 - "FUS-family proteins" should be "FET-family proteins"

Authors' Response: We have changed "FUS-family proteins" to "FET-family proteins" on page 16.

Reviewer Comments: Fig. 5l,m - Which parts of the Raman spectra in k are shown here? Looking at k, it seems that the left side of the spectrum does not go to zero at 1640 cm^{-1} but instead overlaps with another peak. In l,m, the intensity at 1640 cm^{-1} goes to "zero", similar to the right side of the spectrum. Fig. 5l - The third component of the fit (the small peak between 1640 and 1660 cm^{-1}) is hard to see. Is it supposed to represent the nonregular/turns conformations?

Authors' Response: We thank the reviewer for asking this question. In Figure 5l,m, we have zoomed into the amide I region of the Raman spectra shown in Figure 5k which contains a tyrosine band at 1620 cm^{-1} and amide I at 1630-1700 cm^{-1} . In order to better visualize and analyze the amide I band, we have plotted the amide I band that is separately baseline-corrected and deconvoluted. We have now clearly stated this in Supplementary Methods. The third (minor) component of the Raman deconvolution (~1652 cm^{-1}) could potentially represent a minor helical content within wild-type FUS-LC droplets. We have now changed the color scheme for a better visualization of the minor peak.

We thank this reviewer for her/his valuable suggestions to improve our manuscript.